# Multiple nitrogen sources for primary production inferred from $\delta^{13}$C and $\delta^{15}$N in the southern Sea of Japan

Taketoshi Kodama[1,2], Atsushi Nishimoto[3], Ken-ichi. Nakamura[4], Misato Nakae[4], Naoki Iguchi[4], Yosuke Igeta[4], and Yoichi Kogure[4]

[1]Fisheries Resources Institute, Japan Fisheries Research and Education Agency, Yokohama, 236-8648, Japan
[2]Present address: Graduate School of Agriculture and Life Sciences, The University of Tokyo, 113-8657, Japan
[3]Fisheries Technology Institute, Japan Fisheries Research and Education Agency, Yokohama, 236-8648, Japan
[4]Fisheries Resources Institute, Japan Fisheries Research and Education Agency, Niigata, 951-8121, Japan

*Correspondence to*: Taketoshi Kodama (takekodama@g.ecc.u-tokyo.ac.jp)

**Abstract.** Carbon and nitrogen dynamics in the Sea of Japan (SOJ) are rapidly changing. In this study, we investigated the carbon and nitrogen isotope ratios of particulate organic matter ($\delta^{13}C_{POM}$ and $\delta^{15}N_{POM}$, respectively) at depths of $\leq 100$ m in the southern part of the SOJ from 2016 to 2021. $\delta^{13}C_{POM}$ and $\delta^{15}N_{POM}$ exhibited multimodal distributions and were classified into four classes (I–IV) according to the Gaussian mixed model. A majority of the samples were classified as class II ($n = 441$), with mean $\pm$ standard deviation of $\delta^{13}C_{POM}$ and $\delta^{15}N_{POM}$ of $-23.7 \pm 1.2$‰ and $3.1 \pm 1.2$‰, respectively. Compared to class II, class I had significant low $\delta^{15}N_{POM}$ ($-2.1 \pm 0.8$‰, $n = 11$), class III had low $\delta^{13}C_{POM}$ ($-27.1 \pm 1.0$‰, $n = 21$), and class IV had high $\delta^{13}C_{POM}$ ($-20.7 \pm 0.8$‰, $n = 34$). All the class I samples, whose $\delta^{15}N_{POM}$ showed an outlier of total data sets, were collected in winter and had comparable temperature and salinity originating in Japanese local rivers. The generalized linear model demonstrated that the temperature and chlorophyll-*a* concentration had positive effects on $\delta^{13}C_{POM}$, supporting the active photosynthesis and phytoplankton growth increased $\delta^{13}C_{POM}$. However, the fluctuation in $\delta^{15}N_{POM}$ was attributed to the temperature and salinity rather than nitrate concentration, which suggested that the $\delta^{15}N$ of source nitrogen for primary production is different among the water masses. These findings suggest that multiple nitrogen sources, including nitrates from the East China Sea, Kuroshio, and Japanese local rivers, contribute to the primary production in the SOJ.

## 1 Introduction

Carbon and nitrogen dynamics in the oceans are globally changing as a result of anthropogenic activities (Gruber and Galloway, 2008). Carbon and nitrogen isotope values [$\delta^{13}$C and $\delta^{15}$N, the differences from the sample isotope ratios ($^{13}$C/$^{12}$C and $^{15}$N/$^{14}$N) to standard materials] are particularly effective measurements for detecting changes in marine environments (Gruber et al., 1999; Ren et al., 2017) and ecosystems (Lorrain et al., 2020). Earth system model-based techniques have been developed at a global scale in order to comprehend the spatiotemporal variability of carbon and nitrogen isotope values (Buchanan et al., 2019). Marginal seas are strongly affected by anthropogenic activities (Omstedt, 2021). However, because current Earth system models do not focus on them, relevant information is extremely limited.

The Sea of Japan (SOJ) is a western North Pacific semi-closed marginal sea surrounded by the Korean Peninsula, the Japanese Archipelago, and the Russian coast. In the southern part of the SOJ, the Tsushima Warm Current (TWC) flows at the surface

(< 200 m depth) from the west (East China Sea, ECS) to the east (western North Pacific or the Sea of Okhotsk) throughout the year; however, this current is weak in the winter and strong in the summer, and its path exhibits complex variations (Yabe et al., 2021). The TWC governs the elemental cycles and ecosystem in the southern part of the SOJ (Kodama, 2020), and its water is a mixture of the Kuroshio Current, the Taiwan Warm Current, and Changjiang discharged waters (Isobe, 1999; Guo et al., 2006). The elemental cycles in the surface layer of the SOJ are rapidly changing, and the pH, phosphate, and oxygen concentrations have been decreasing over the past five decades (Ono, 2021; Kodama et al., 2016; Ishizu et al., 2019).

Studies in the SOJ have been undertaken using a stable isotope ratio of zooplankton and small pelagic fish tissues to determine changes in carbon and nitrogen dynamics (Nakamura et al., 2022; Ohshimo et al., 2021). These studies found that the $\delta^{13}C$ in animal tissues declined at the same rate as the Suess effect, but there was no significant linear trend in the $\delta^{15}N$ (Nakamura et al., 2022; Ohshimo et al., 2021). The $\delta^{15}N$ in animal tissues varies with trophic position. In the SOJ, the zooplankton biomass has been found to be negatively coupled with small pelagic fish biomass over the last half-century, and hence, food-web structure may periodically change (Kodama et al., 2022a). Thus, the baseline $\delta^{15}N$ in marine ecosystems is vital for identifying changes in nitrogen dynamics in the SOJ. However, reports on the variation in stable isotope ratios of particulate organic matter (POM) in the surface layer, which are the baseline values of marine ecosystems, are confined to the Japanese coastal areas (Antonio et al., 2012; Nakamura et al., 2022). As a result, the variabilities in POM stable isotope ratios with regard to environmental parameters are not clearly understood.

Two recent studies have indicated that the carbon and nitrogen stable isotope ratio of POM varies in the western North Pacific marginal seas (Ho et al., 2021; Kodama et al., 2021). The $\delta^{13}C$ of POM increased as the phytoplankton abundance increased in the southwestern ECS, which is consistent with isotope fractionation that occurs during photosynthesis (Ho et al., 2021). The $\delta^{15}N$ of this area decreased with freshwater intake and coastal upwelling in the summer, and was negatively correlated with nitrate concentration (Ho et al., 2021). Gao et al. (2014) reported the horizontal distribution of $\delta^{13}C$ and $\delta^{15}N$, and that $\delta^{13}C$ largely varies in this sea. Active nitrogen fixation decreases the $\delta^{15}N$ of POM in offshore waters in the Kuroshio area during the summer, whereas the $\delta^{15}N$ of POM remains high in the coastal waters (Kodama et al., 2021).

The $\delta^{13}C$ and $\delta^{15}N$ of POM are thought to be more changeable in the SOJ than in the other western North Pacific marginal seas. In the case of carbon, Kosugi et al. (2016) reported that the partial pressure of $CO_2$ ($pCO_2$) on the surface of the central and eastern parts of the SOJ (312–329 µatm) differs from that in the northwestern region (360–380 µatm). Furthermore, the carbon:nitrogen ratio (C:N ratio) of POM in the surface water of the northern ECS is extremely high (>40:1) near Japan (Gao et al., 2014), and this organic carbon-rich water may influence the carbon dynamics of the SOJ. In the case of nitrogen, deep mixing occurs in the SOJ in winter (Ohishi et al., 2019); thus, deep-seawater-originated nitrate contributes to the primary production in the SOJ. The previous nutrient dynamics studies have revealed, however, that the nitrate in the SOJ is supplied from the Kuroshio, regeneration at the bottom of the ECS, the Changjiang diluted waters, and the atmosphere in the summer (Kim et al., 2011; Kodama et al., 2017; Kodama et al., 2015). In addition, diazotrophs are present during the summer (Hashimoto et al., 2012; Sato et al., 2021). These "new nitrogen" contributions to the primary production are not evaluated in the SOJ. The primary production in the northeastern ECS is considered to be supported by a variety of nitrogen sources, such

as atmospheric deposition, Changjiang River discharge, and Kuroshio waters with varying $\delta^{15}N$ (Umezawa et al., 2021; Umezawa et al., 2014). The decrease in phosphate concentration in the SOJ is mainly observed with an increase in the nitrogen supply in the ECS (Kodama et al., 2016; Kim et al., 2013). Thus, the high contribution of atmospheric deposition, Changjiang River discharge, and Kuroshio waters is expected in the SOJ as well as the northeastern ECS, and the $\delta^{15}N$ of POM helps in identifying which nitrogen source mainly supports primary production in this sea. Therefore, in this study, we investigated the carbon and nitrogen stable isotope ratios in the southern SOJ (approximately $\leq 40°N$). The main objectives were to: 1) demonstrate the spatial distribution of carbon and nitrogen stable isotope ratios, and 2) identify which nitrogen source mainly supports the primary production.

## 2 Materials and Methods

### 2.1 Sampling

POM samples with environmental data were collected from 2016 to 2021 in the southern part of the SOJ between 35°N and 41°N and between 131°15'E and 139°50'E (Fig. 1) during 26 cruises (Table 1). The cruises were held yearly from February to September, except for March (Table 1, Fig. 1). The observations were repeated 15 times after 2016 along the monitoring line established off the Sado Island (SI-line, Table 1, Fig. 1).

POM samples were generally obtained at the depths of 10 and 30 m, or at the subsurface chlorophyll-*a* maximum (SCM). The vertical profiles of temperature, salinity, and chlorophyll-*a* fluorescence were monitored in real-time using a conductivity-temperature-depth (CTD) sensor (SBE9plus, Seabird Electronics). Real-time observations were not performed in some specific cases; here, the record-type CTD (SBE19, Seabird Electronics) without a fluorometer was used instead of the SBE9plus. On this basis, we set 30 m as the representative subsurface layer for the spring (March–June) and summer (July–September) seasons as it is below the surface mixed layer in SOJ. During the cruise in September 2020, the POM samples were collected at six depths (0, 10, 30, 50, and 100 m, and SCM) to determine the variations between the sampling layers. In the July and August cruises, photosynthetically active radiation (PAR) was occasionally accessible using a PAR sensor (Seabird Electronics) attached to the CTD. In the summer of SOJ, the mean ($\pm$ SD) euphotic zone depth—where PAR is 1% of the surface—was 50 ($\pm$ 14) m ($n = 207$), and the PAR at a depth of 30 m was $6.8 \pm 4.3\%$ of the surface. In May and August 2020, we collected samples at depths of 75 m ($n = 3$) and 100 m ($n = 1$), respectively. In spring and summer, considering the mixed layer and the euphotic zone depth, the layers at 0–10 m, 20–65 m, and 75–100 m depths were defined as the surface, subsurface, and deep layers, respectively, to detect the influence of the vertical difference of unevaluated characteristics such as light and phytoplankton community. In winter, all the samples were considered as surface layer samples because the difference in potential density between 10 and 30 m depths was <0.125 at every station, suggesting that they were within the mixed layer. Temperature, salinity, nutrients (nitrate, nitrite, silicate, and phosphate), and chlorophyll-*a* concentrations were collected and utilized as environmental parameters at the same depth as the POM samples. Water samples were obtained using Niskin bottles mounted on a carousel or a bucket to measure nutrient, chlorophyll-*a*, and POM concentrations. The nutrient and chlorophyll-

*a* concentrations were analyzed following Kodama et al. (2015). In this study, we only used nitrate concentration, because the

silicate and phosphate concentrations showed the similar variation with nitrate concentration.  The detection limits of the nitrate concentrations were 0.01–<0.1 µM according to the standard deviations of blank values during the measurements. To treat the common-logarithm transformed values, the nitrate concentrations of <0.01 µM (below the detection limit) were set as 0.01 µM. The temperature and salinity at the 0-m depth were defined as the values observed by the CTD sensors at 1-m depth because salinity at the 0-m depth sometimes exhibited "unreliable" values.

**2.2 Isotope analyses**

To measure the mass and the carbon and nitrogen stable isotope ratios of the POM, 0.9–11.5 L of seawater was filtered using a pre-combusted (450°C, 6 h) glass fiber filter (pore size: 0.7 µm, GF/F, Whatman). The filtration was stopped in 2 h after sampling, and > 5 L seawater was filtered in 2 h for most of the samples. After the shipboard filtration, the samples were frozen (<–20°C). Onshore laboratory analyses were performed using methods from Kodama et al. (2021), whose preservation differed

from that of Lorrain et al. (2003), while the other protocols remained the same. The masses of $^{12}$C, $^{13}$C, $^{14}$N and $^{15}$N were determined using the same sample. The samples were exposed to HCl fumes for > 2 h to remove carbonate salt, dried in an oven (60°C) overnight and stored in a desiccator until isotope ratio measurements. After another round of oven-drying, the entire glass filter was wrapped in a tin cup. Subsequently, the carbon and nitrogen isotope ratios were measured using an Isoprime 100 isotope ratio mass spectrometer (Elementar, Langenselbold, Germany). In 2016, the colored surfaces of the glass

filters were scraped off, and the $\delta^{13}$C and $\delta^{15}$N were measured. The absolute amounts of carbon and nitrogen in these scraped samples were different from those in the whole samples. The $\delta^{13}$C and $\delta^{15}$N were calibrated using curves obtained from L-alanine (Shoko Science) during the measurements. The $\delta^{13}$C and $\delta^{15}$N of L-alanine were determined by Shoko Science and considered as the reference materials (Sato et al., 2014). The $\delta^{13}$C and $\delta^{15}$N of POM sample ($\delta^{13}$C$_{POM}$ and $\delta^{15}$N$_{POM}$, respectively) were expressed using Eq. (1):

$$\delta^{13}C_{POM} \text{ or } \delta^{15}N_{POM} = (R_{sample}/R_{reference} - 1.0) \times 1000, \tag{1}$$

where $R_{sample}$, and $R_{reference}$ are the heavy ($^{13}$C and $^{15}$N) to light ($^{12}$C and $^{14}$N) isotope ratios of the POM sample and reference, respectively. The reference materials were atmospheric N$_2$ for nitrogen and Vienna Pee Dee Belemnite for carbon. Given the quality of the L-alanine, and $\delta^{13}$C and $\delta^{15}$N were rounded off to one decimal place, the precision of the analyses was within 0.2‰.

The C:N ratio of POM occasionally exhibited outliers, due to which, the mean and standard deviations (SD) of the C:N ratio were calculated and the samples whose C:N ratio differed by > 3 × SD from the mean value were eliminated. After elimination, we re-calculated the mean and SD of the C:N ratio, and the samples whose C:N ratio differed by > 3 × recalculated SD from the recalculated mean value were eliminated. As a result, six samples were removed. Then, the nine samples lacking environmental data were removed. Therefore, 507 samples were used in this study. Among these samples, 101 samples were

obtained along the SI-line.

## 2.3 Statistical analyses

All statistical analyses were conducted using R (R Core Team, 2023). Accompanied by the analysis of variance (ANOVA), a pairwise test with the *Tukey–Kramer* adjacent was applied to the least-squared mean (lsmean) values. The "ggeffect" package (Lüdecke, 2018) was used to calculate the lsmean values and SEs. The "car" package (Fox and Weisberg, 2018) of ANOVA was used to conduct type II ANOVA for unbalanced data comparison. Neither $\delta^{13}C_{POM}$ nor $\delta^{15}N_{POM}$ exhibited a normal distribution according to the *Kolmogorov–Smirnov* test ($p < 0.001$); both showed multimodal distributions. The linear models cannot be applied to the multimodal distributions, we applied a two-dimensional Gaussian mixed model (GMM) to classify $\delta^{13}C_{POM}$ and $\delta^{15}N_{POM}$ using the "mclust" package (Scrucca et al., 2016). The samples were not divided based on the sampling depth and seasons. The Bayesian information criterion (BIC) determined the number of classes.

Generalized linear models (GLMs) were applied to assess the variables that predicted the relationships between the environmental parameters and $\delta^{13}C_{POM}$ and $\delta^{15}N_{POM}$ as in Kodama et al. (2021). We assumed that the error distributions of $\delta^{13}C_{POM}$ and $\delta15N_{POM}$ were normally distributed with a linear link function in the GLMs. The full GLM models are as follows:

$$\delta X_{POM} \sim glm(f(class) + f(layer) + f(season) + Lon + Lat + C/N + T + S + Chl + Nit] \tag{2}$$

where $\delta X_{POM}$, Lon, Lat, C/N, T, S, Chl, and Nit denote $\delta^{13}C_{POM}$ or $\delta^{15}N_{POM}$, longitude, latitude, C:N ratio, temperature, salinity, chlorophyll-*a* concentration, and nitrate concentration, respectively. The chlorophyll-*a* and nitrate concentrations were transformed into logarithmic values. The arguments of the *f* functions are categorical variables that are used to simulate non-linear relationships. The numbers of classes were defined by GMM and BIC, while those of layer and season were three each (surface, subsurface, and 100-m depth in layer and winter [January–February], spring [March–June], and summer [July–September] in season). The explanatory variables used in the full GLMs were chosen based on the retraction of multicollinearity. We also used GLM approaches that included a quadratic expression in the model, as Nakamura et al. (2022) did, as well as a generalized additive model approach, as Kodama et al. (2021) did. However, as the deviance-explained values of the models did not improve, we opted for the simple GLM approach.

The explanatory variables and final GLM descriptions were selected using the corrected Akaike information criterion (AIC), which may be used to assess the likelihood of the model. The lsmean values and SEs based on the AIC-selected GLMs were used to visualize the influence of the explanatory variables on the $\delta^{13}C_{POM}$ or $\delta^{15}N_{POM}$. ANOVA was used to test the effects of the lsmean values of the categorical variables (the class, season, and depth when they remained).

We did not include the interaction terms in the GLMs. For instance, longitude–latitude interactions can reflect the 2-D variations. Moreover, the temperature–salinity (T–S) diagram is the most fundamental method for obtaining the characteristics of water masses. Therefore, although the interactions must be considered, the interaction terms of temperature and salinity could not be used instead of the T–S diagram in the GLMs.

## 3 Results

### 3.1 Spatial distribution of $\delta^{13}C_{POM}$ and $\delta^{15}N_{POM}$

The $\delta^{13}C_{POM}$ and $\delta^{15}N_{POM}$ varied from –29.3 to –17.7‰ and –3.2 to 6.7‰, respectively. In September 2020, the vertical profiles (0, 10, 30, 50, and 100 m depths and SCM [34–56 m]) of $\delta^{13}C_{POM}$ and $\delta^{15}N_{POM}$ were collected at nine stations. The differences in the $\delta^{13}C_{POM}$, $\delta^{15}N_{POM}$, and C:N ratios were significant among the layers according to ANOVA ($p < 0.001$). The $\delta^{13}C_{POM}$ was the lowest at a depth of 50 m or SCM (mean ± SD: -25.7 ± 0.4‰ and –25.4 ± 0.5‰, respectively, Fig. 2a). The $\delta^{13}C_{POM}$ decreased with a depth of up to 50 m (or SCM layer) and slightly increased at the 100-m depth (–24.2 ± 1.3‰). The $\delta^{15}N_{POM}$ was the lowest at 10-m depth (1.5 ± 0.6‰), increased with depth, and the highest value was identified at the 100-m depth (5.0 ± 0.6‰) (Fig. 2b). The mean C:N ratio was estimated to 5.4–6.3 mol mol$^{-1}$ except the 100-m depth with 4.2 ± 0.69 mol mol$^{-1}$ (Fig. 2c). When the subsamples at the nine stations were regrouped into the surface (0 and 10 m depths) and subsurface (30 and 50 m and SCM) groups, the differences in both $\delta^{13}C_{POM}$ and $\delta^{15}N_{POM}$ were significant between the surface and subsurface (*t*-test, $p \leq 0.018$). Therefore, $\delta^{13}C_{POM}$ and $\delta^{15}N_{POM}$ were different between the surface and subsurface, particularly during summer.

The horizontal variations in $\delta^{13}C_{POM}$ are shown by the 1° × 1° grid median values (Fig. 3a–e). During winter, the median (± interquartile range, IQR) of the $\delta^{13}C_{POM}$ was –25.0 ± 1.6‰, and low $\delta^{13}C_{POM}$ was observed in the offshore waters. During spring, low $\delta^{13}C_{POM}$ was identified offshore, and the surface $\delta^{13}C_{POM}$ (–24.6 ± 1.6‰) was lower than that of the subsurface (–23.7 ± 1.4‰, *Wilcoxon* test, W = 271, $p = 0.021$). During summer, the surface $\delta^{13}C_{POM}$ (–22.9 ± 1.0‰) was significantly higher than that of the subsurface (–24.4 ± 1.9‰, *Wilcoxon* test, W = 36002, $p < 0.001$). The $\delta^{13}C_{POM}$ in the subsurface during the summer was higher in the western than in the eastern part, while those of the surface were higher in the eastern than in the western part.

The horizontal variations in $\delta^{15}N_{POM}$ are shown in the same manner as $\delta^{13}C_{POM}$ (Fig. 3f–j). Evidently, in winter, lower values (~ –2.0‰) were observed. $\delta^{15}N_{POM}$ was higher in spring (3.1 ± 1.8‰ and 3.2 ± 0.8‰ in the surface and subsurface, respectively) than in winter (-2.0 ± 2.0‰). Moreover, during spring, the $\delta^{15}N_{POM}$ value in the offshore area was lower than that of the coastal area. In summer, the surface $\delta^{15}N_{POM}$ (2.8 ± 1.7‰) was significantly lower than that of the subsurface (3.1 ± 1.5‰, *Wilcoxon* test, W = 19187, $p = 0.02874$). Furthermore, lower $\delta^{15}N_{POM}$ values were identified in the offshore water, and high $\delta^{15}N_{POM}$ values were observed in the coastal area of the northern part.

### 3.2 Temporal variations of $\delta^{13}C_{POM}$ and $\delta^{15}N_{POM}$ along the SI-line

The monthly variations in $\delta^{13}C_{POM}$ and $\delta^{15}N_{POM}$ were evaluated using the samples collected along the SI-line (Fig. 4a–d). When the interannual variations were ignored, $\delta^{13}C_{POM}$ of the surface was the lowest in April (lsmean ± SE: –25.7 ± 0.5‰, $n = 8$) and highest in September (–22.3 ± 0.33‰, $n = 23$) (Fig. 4a); no significant difference was observed during February, April, and June (pairwise test with *Tukey–Kramer* adjacent, $p > 0.05$). In the subsurface layer, the monthly variation of $\delta^{13}C_{POM}$ was not significant (ANOVA, $p = 0.09$) (Fig. 4b). The $\delta^{15}N_{POM}$ of the surface layer was the lowest in February (–1.1 ± 0.4‰, $n = $

15) and highest in July (3.1 ± 0.5‰, $n = 7$), while a significant difference was observed only observed in February and other months (Fig. 4b). In the subsurface layer, the monthly variation of $\delta^{13}C_{POM}$ was not significant (ANOVA, $p = 0.09$) (Fig. 4c), and that of $\delta^{15}N_{POM}$ was significant ($p = 0.033$) and a significant difference ($p = 0.025$) was observed between April (1.0 ± 0.6‰, $n = 3$) and June (3.3 ± 0.4‰, $n = 6$, Fig. 4d).

### 3.3 Classifications of $\delta^{13}C_{POM}$ and $\delta^{15}N_{POM}$

The samples were divided into four classes (I–IV) based on $\delta^{13}C_{POM}$ and $\delta^{15}N_{POM}$ according to the two-dimensional GMM (Fig. 5). The mean ± SDs of $\delta^{13}C_{POM}$ and $\delta^{15}N_{POM}$ were –24.6 ± 1.2‰ and –2.1 ± 0.8‰ in class I ($n = 11$), –23.7 ± 1.2‰ and 3.1 ± 1.2‰ in class II ($n = 441$), –27.1 ± 1.0‰ and 2.0 ± 1.3‰ in class III ($n = 21$), and –20.7 ± 0.8‰ and 1.7 ± 1.0‰ in class IV ($n = 34$), respectively (Fig. 5b–c). The $\delta^{13}C_{POM}$ values were significantly different among the different classes ($p < 0.001$, pairwise test with *Tukey–Kramer* adjacent), except between classes I and II ($p = 0.06$) (Fig. 5b). The $\delta^{15}N_{POM}$ values were significantly different among the different classes ($p < 0.001$, pairwise test with *Tukey–Kramer* adjacent), except between classes III and IV ($p = 0.76$) (Fig. 5c). However, the mean ± SD values of $\delta^{15}N_{POM}$ in classes III and IV were within the range of class II values (Fig. 5c). Therefore, the characteristics of $\delta^{13}C_{POM}$ and $\delta^{15}N_{POM}$ in classes I, II, III, and IV were middle-$\delta^{13}C_{POM}$ and low-$\delta^{15}N_{POM}$, middle-$\delta^{13}C_{POM}$ and high-$\delta^{15}N_{POM}$, low-$\delta^{13}C_{POM}$ and high-$\delta^{15}N_{POM}$, and high-$\delta^{13}C_{POM}$ and high-$\delta^{15}N_{POM}$, respectively.

Environmental conditions (temperature, salinity, nitrate concentration, chlorophyll-*a* concentration, and C:N ratio) differed significantly among the classes (ANOVA, $p < 0.01$; Fig. 6). The temperature was lower in class I (mean ± SD: 10.1 ± 2.0°C) and III (12.7 ± 6.0°C) than in classes II (20.1 ± 4.9°C) and IV (24.0 ± 2.6°C) (Fig. 6a). Significant differences in temperature were discerned ($p < 0.003$, pairwise test with *Tukey–Kramer* adjacent) among different classes, except between classes I and III ($p = 0.44$). The only significant difference in salinity among the classes was discerned between classes II (33.76 ± 0.79) and III (34.27 ± 0.18) ($p = 0.01$, pairwise test with *Tukey–Kramer* adjacent) (Fig. 6b). The mean salinity with SD of classes I and IV were 33.99 ± 0.06 and 33.91 ± 0.36, respectively (Fig. 6b). The mean nitrate concentration values were high in classes I (4.20 ± 1.31 μM) and III (3.88 ± 3.86 μM), lower in class II (0.51 ± 1.31 μM), and the lowest in class IV (0.05 ± 0.01 μM) (Fig. 6c). The chlorophyll-*a* concentrations were significantly different among the classes (ANOVA, degrees of freedom [DF] = 3, $F$-value = 2.643, $p = 0.048$), but no significant differences were identified among the pairs ($p > 0.058$). The chlorophyll-*a* concentration was the highest in class IV (0.69 ± 1.33 μg L$^{-1}$), followed by class I (0.61 ± 0.23 μg L$^{-1}$), class III (0.58 ± 0.37 μg L$^{-1}$), and the lowest in class II (0.44 ± 0.48 μg L$^{-1}$) (Fig. 6d). Ahigh C:N ratio was identified in class IV (7.06 ± 1.50 mol mol$^{-1}$), which was significantly higher ($p < 0.01$, pairwise test with *Tukey–Kramer* adjacent) than in classes II (6.13 ± 1.39 mol mol$^{-1}$) and III (5.12 ± 1.22 mol mol$^{-1}$) while the C:N ratio in class IV did not differ from that in class I (6.05 ± 0.58 mol mol$^{-1}$) (Fig. 6e).

The T–S diagram demonstrated that all the samples of class I were observed in winter, and mostly (10 of 11 samples) in the water with the temperature and salinity within the 9.4–11.4°C and 33.877–34.038 ranges, respectively (Fig. 7). Only a single sample of class I was identified at a temperature and salinity of 4.3°C and 34.037, respectively (Fig. 7). The class III samples

were never observed >24°C, and twenty of the twenty-one samples were observed <19°C (Fig. 6), but did not group into similar T-S characters (Fig. 7). The class IV samples were only observed during summer, plotted in the warm water mass in summer of 2019, but also observed in the different T–S characters in the other year (Fig. 7).

### 3.4 Relationships with environmental conditions

According to the GMM, both $\delta^{13}C_{POM}$, and $\delta^{15}N_{POM}$ exhibited multi-modal distribution. The linear regression was based on the assumption that the dependent variable has a normal distribution; therefore, in this study, the linear regressions were inappropriate for evaluating the relationships. We have provided the results of the linear regression analysis in the supporting information (Fig. S1).

The least AIC GLM for $\delta^{13}C_{POM}$ was as follows:

$\delta^{13}C_{POM} \sim glm[f(class) + f(layer) + Lat + T + S + Chl + Nitrate]$ (3).

The $r^2$ value of the least AIC $\delta^{13}C_{POM}$ model was found to be 0.659. ANOVA indicated that all the remaining explanatory variables were significant ($\chi^2 \geq 7.53, p \leq 0.006$). The responses of latitude, salinity, and nitrate concentration were significantly negative ($p < 0.001$), whereas those of temperature and chlorophyll-$a$ concentration were significantly positive ($p < 0.001$) (Fig. 8). The lsmean $\delta^{13}C_{POM}$ with ANOVA suggested that classes III (lsmean ± SE: –25.6 ± 0.3‰) and IV (–20.6 ± 0.2‰)

were significantly higher and lower than those of other classes, respectively (pair-wise test with *Tukey–Kramer* adjacent, $p <$ 0.001). The difference between classes I (–23.0 ± 0.4‰) and II (–23.4 ± 0.1‰) was insignificant (pair-wise test with *Tukey–Kramer* adjacent, $p = 0.542$) (Fig. 8a). The $\delta^{13}C_{POM}$ of the surface (lsmean ± SE: –23.4 ± 0.1‰) was higher than that of the subsurface (–23.7 ± 0.1‰, pair-wise test with Tukey's adjacent, $p = 0.0154$). Moreover, $\delta^{13}C_{POM}$ at the 100-m depth (–22.4 ± 0.3‰) was significantly higher than those of the surface and subsurface ($p \leq 0.001$) (Fig. 8).

The least-AIC $\delta^{15}N_{POM}$ GLM was as follows:

$\delta^{15}N_{POM} \sim glm[f(class) + f(season) + f(layer) + Lon + Lat + T + S]$ (4).

All the parameters were statistically significant (ANOVA, $p < 0.001$), and the $r^2$ value of the model was found to be 0.451. The longitude had a positive impact on $\delta^{15}N_{POM}$, i.e., $\delta^{15}N_{POM}$ increased eastward (Fig. 9e). The $\delta^{15}N_{POM}$ was negatively affected by the latitude, temperature, and salinity (Fig. 9d, f, g). The lsmean values indicated that the $\delta^{15}N_{POM}$ of classes I

(lsmean ± SE: –1.6 ± 0.4‰) and II (3.1 ± 0.2‰) were significantly lower and higher than those of the other three classes, respectively (pair-wise test with *Tukey–Kramer* adjacent, $p < 0.01$). Moreover, classes III (2.1 ± 0.3‰) and IV (2.0 ± 0.3‰) were not significantly different ($p = 0.9735$). The seasonal variations exhibited significant differences between spring (1.3 ± 0.2‰) and summer (1.9 ± 0.2‰, pair-wise test with *Tukey–Kramer* adjacent, $p < 0.01$), while the lowest lsmean values were recorded in winter (0.9 ± 0.4‰). No significant differences of lsmean $\delta^{15}N_{POM}$ were found between spring and summer ($p >$

0.0897). The layer indicated that $\delta^{15}N_{POM}$ at a depth of 100 m (2.3 ± 0.4‰) was significantly higher (pair-wise test with *Tukey–Kramer* adjacent, $p < 0.01$) than that at the surface (0.9 ± 0.2‰) and subsurface (1.0 ± 0.2‰).

## 4. Discussion

This study, for the first time, revealed $\delta^{13}C_{POM}$ and $\delta^{15}N_{POM}$ in a wide area of the southern SOJ. Previous studies on $\delta^{13}C_{POM}$ and $\delta^{15}N_{POM}$ were confined to the coastal areas of the southern SOJ (Antonio et al., 2012; Nakamura et al., 2022) and sinking particles (Nakanishi and Minagawa, 2003). Seasonality was demonstrated by the $\delta^{13}C$ and $\delta^{15}N$ values of sinking organic particles in the deep layer (> 500 m depth) (Nakanishi and Minagawa, 2003). Significant seasonality of $\delta^{13}C_{POM}$ and $\delta^{15}N_{POM}$ was observed on the surface in our study, although the pattern of $\delta^{13}C$ differed from that seen by Nakanishi and Minagawa (2003). In Nakanishi and Minagawa (2003), the $\delta^{13}C$ of sinking particles increased with the bloom period. However, in our study, the $\delta^{13}C_{POM}$ decreased along the SI-line. On the contrary, the seasonality of $\delta^{15}N$ of the sinking particles and $\delta^{15}N_{POM}$ was comparable. We were unable to determine why the seasonality of $\delta^{13}C_{POM}$ of our study differed from that of Nakanishi and Minagawa (2003). However, our observations indicated that the $\delta^{13}C_{POM}$ and $\delta^{15}N_{POM}$ values at 100-m depth, below the euphotic layer, were significantly different from those in the euphotic zone (Fig. 2), and monthly variations were not significant below the mixed layer. Thus, the characteristics of the sinking particles in the SOJ may be different from those of POM in the surface layer.

### 4.1. Variations and the causes of carbon isotope ratio

According to the GLM approach, environmental characteristics may explain approximately two-thirds of the variations in $\delta^{13}C_{POM}$. The relationships found in this study are consistent with previous observations. When environmental conditions were ignored, seasonal variation was significant; nevertheless, seasonal variation was not selected as an explanatory variable in the GLM, suggesting that environmental parameters can well explain the seasonality of $\delta^{13}C_{POM}$. The positive associations observed between temperature or chlorophyll-*a* concentration and $\delta^{13}C_{POM}$ were similar to those observed in previous ocean and incubation experiments (Fontugne and Duplessy, 1981; Goericke and Fry, 1994; Miller et al., 2013; Savoye et al., 2003). Phytoplankton community growth rates are probably high in the warm water (Sherman et al., 2016), and high chlorophyll-*a* concentrations are considered to be a consequence of rapid phytoplankton growth. During the rapid growth phase, phytoplankton utilizes more $^{13}CO_2$ in the water, consequently elevating $\delta^{13}C_{POM}$ (Freeman and Hayes, 1992). In addition, the C:N ratio was not selected in the GLM but had a positive relationship with $\delta^{13}C_{POM}$ (Figure S1). Tanioka and Matsumoto (2020) reported that C:N ratio elevates with the increase of light based on the meta-analysis, supporting the active photosynthesis elevates both C:N ratio and $\delta^{13}C_{POM}$.

The negative relationship between $\delta^{13}C_{POM}$ and salinity or nitrate concentration has been previously reported in Kuroshio and western North Pacific boundary currents (Kodama et al., 2021). The negative relationship between $\delta^{13}C_{POM}$ and salinity in the Kuroshio area was considered to be a mixture of POM formed in the estuary (Kodama et al., 2021), where phytoplankton bloom occurs and then high $\delta^{13}C_{POM}$ POM are formed (Savoye et al., 2003; Ogawa and Ogura, 1997). Although our samples were occasionally obtained near the Japanese coast, the less saline water of this sea is mostly due to the influence of the ECS (Kosugi et al., 2021). In the ECS, $\delta^{13}C_{POM}$ is negatively associated with salinity (Ho et al., 2021), and high $\delta^{13}C_{POM}$ (> –20‰)

is detected in the Changjiang diluted waters due to the sediment resuspension in the Changjiang estuary (Gao et al., 2014), while the relationship between salinity and $\delta^{13}C_{POM}$ was unknown in Gao et al. (2014) because they did not report the hydrographic characteristics in the high $\delta^{13}C_{POM}$ area. These facts indicated that the negative relationship between $\delta^{13}C_{POM}$ and salinity pointed to the importance and impact of the processes at the Changjiang estuary on the SOJ. The direct influence of terrestrial organic matter from China may be ignored: the $\delta^{13}C_{POM}$ in the Changjiang is $< -25\permil$ (Gao et al., 2014), while the contribution of terrestrial organic matter is <10% 500 km away from the Changjiang estuary (Wu et al., 2003). The effects of Japanese local rivers cannot be ignored, where river-originated $\delta^{13}C_{POM}$ is $\sim -24\permil$ and lower than the ocean-originated $\delta^{13}C_{POM}$ (Antonio et al., 2012), and thus, the direct influence of terrestrial organic matter is deemed to be restricted in the SOJ based on the relationship between salinity and $\delta^{13}C_{POM}$. These results suggest that POM with high-$\delta^{13}C_{POM}$ from the less saline waters of the ECS is transported into the SOJ and influences the spatiotemporal variation of $\delta^{13}C_{POM}$ in the SOJ, particularly during the summer, when the Changjiang-origin freshwater inputs to the SOJ via the Tsushima Strait are the highest among the seasons (Morimoto et al., 2009). We could not come up with a plausible explanation for the negative relationship between latitude and nitrate concentration.

The environmental parameters properly explained $\delta^{13}C_{POM}$ variability, but not the difference among classes (III and IV) because it was significant in the GLM. Temperature, nitrate concentration, and C:N ratio were found to be significantly different across classes III and IV. The high $\delta^{13}C_{POM}$ with low nitrate concentration and high C:N ratio in the class IV samples may be attributed to active carbon assimilation under nitrate depletion conditions, while the chlorophyll *a* concentration was low in the class IV samples. In contrast to class IV samples, low $\delta^{13}C_{POM}$ samples in class III were mostly observed in nitrate-rich waters, indicating that primary production is not active due to light limitation induced by the deep mixing and sampling below the light-compensation depth; the previous study's iron limitation for primary production was rejected (Fujita et al., 2010). Liu et al. (2007) reported that low $\delta^{13}C_{POM}$ is observed at the subsurface layer in the South China Sea due to low growth rate, which corresponds to our results. However, comparable environmental conditions of classes III and IV samples were observed in the classes I and II samples, and hence the fundamental reason why the $\delta^{13}C_{POM}$ was low and high in classes III and IV, respectively, is yet unclear. According to the T-S diagram, class IV POM was in the warm and saline waters in 2019. This may indicate that the isotope fraction in this water is different; for example, an increase in diatom abundance elevates $\delta^{13}C_{POM}$ (Lowe et al., 2014), but the diatom contribution in the SOJ is low during the summer (Kodama et al., 2022b). The phytoplankton community structure was not assessed in this study and will need to be investigated more in the future.

### 4.2. Unique nitrogen dynamics in the SOJ

Unlike $\delta^{13}C_{POM}$, $\delta^{15}N_{POM}$ was not adequately explained by environmental parameters based on the detection coefficient values. Temperature and salinity remained as explanatory variables in the variation of $\delta^{15}N_{POM}$ in the ocean, but were not regarded as essential determinants (Sigman et al., 2009). Temperature had the opposite impact reported in the Kuroshio (Kodama et al., 2021). Temperature had no significant effect in the western south ECS (Ho et al., 2021), thereby suggesting that the negative impact of temperature on the $\delta^{15}N_{POM}$ was specific to the SOJ; however, its mechanisms remain unclear.

The negative impact of salinity was similar to that observed in the Kuroshio (Kodama et al., 2021). In June 2010, the $\delta^{15}N_{POM}$ in the surface less-saline Changjiang diluted water (at 5 m depth and salinity <30) was recorded to be ~9‰ in the northern part of the ECS (Sukigara et al., 2017). In 2011, the $\delta^{15}N_{POM}$ in the Changjiang diluted water in 2011 was ~6‰ (Sukigara et al., 2017). Sukigara et al. (2017) stated that POM with high $\delta^{15}N_{POM}$ was not always present in the Changjiang diluted water, although high $\delta^{15}N_{POM}$ was also reported in the ECS in previous studies (Gao et al., 2014; Wu et al., 2003). These findings corroborate the hypothesis that high $\delta^{15}N_{POM}$ levels originate from the less-saline Changjiang diluted water that mainly flows into the SOJ during the summer (Morimoto et al., 2009), as well as the negative relationship between salinity and $\delta^{15}N_{POM}$.

There are two proposed mechanisms for the influence of latitude on the $\delta^{15}N_{POM}$. One is the Japanese territorial influence. The other is the impact of the coastal branch of the Tsushima Warm Current. The coastal branch of the Tsushima Warm Current originates from the eastern channel of the Tsushima Strait and flows along the Japanese coast (Katoh, 1994). Our monitoring regions always included a coastal region in the low latitude area, as well as flows of the coastal branch of the Tsushima Warm Current (Katoh, 1994). The $\delta^{13}C_{POM}$ did not indicate direct territorial inputs in this case, thus the influence of the coastal branch of the Tsushima Warm Current and the origin of the waters may be the explanation.

Although ocean observations indicated a negative relationship between $\delta^{15}N_{POM}$ and nitrate concentration based on Rayleigh fractionation (Kodama et al., 2021; Ho et al., 2021), the negative relationship in the SOJ remained equivocal. Even in an open system, kinetic isotope effects ($\delta^{15}N$ difference between reactant and product) and the degree of consumption of reactant theoretically determine the $\delta^{15}N$ value of the product, and when the remaining reactant is zero (i.e., completely consumed), the $\delta^{15}N$ of a product is equal to the original $\delta^{15}N$ of reactant (Sigman et al., 2009). In this case, the kinetic isotope effect of nitrate on POM is ~3‰ (Sigman et al., 2009). Thus, $\delta^{15}N_{POM}$ is theoretically 0–3‰ lower than $\delta^{15}N$ of nitrate ($\delta^{15}N_{NO3}$), and with nitrate consumption, it increases and approaches the original $\delta^{15}N_{NO3}$ value. The monthly variations in the surface layer along the SI-line indicated that the $\delta^{15}N_{POM}$ increases from winter to summer, and the GLM method confirms this trend. This also indicates that nitrate depletion partly contributes to the increase in $\delta^{15}N_{POM}$. Because $\delta^{15}N_{POM}$ was not normally distributed, the association was insignificant ($p = 0.4467$) when we removed class I POM, indicating that the relationships between $\delta^{15}N_{POM}$ and the environment, especially the nitrate concentration in the SOJ, were unique.

The following are two hypotheses for the ambiguous relationship between $\delta^{15}N_{POM}$ and nitrate concentration: (1) our dataset; and (2) the variety of nitrogen sources. First, our observations were mainly conducted in the summer, during which time the nitrate was depleted at the surface, and the nitrate concentration in 182 of the 494 samples was not detectable (< 0.01 μM). The $\delta^{15}N_{POM}$ in the nitrate-depleted waters varied greatly (mean ± SD: 2.8 ± 1.2‰, $n = 182$). This $\delta^{15}N_{POM}$ variation in nitrate-depleted water may have rendered the relationship between $\delta^{15}N_{POM}$ and nitrate concentration unclear, rendering the association statistically insignificant. When the GLM approach was conducted for subsamples with detectable nitrate (>0.01 μM), the nitrate concentration remained as the explanatory variable in the least-AIC model; the coefficient was negative but not significant (ANOVA, $p = 0.12$). As a result, the imbalanced dataset was not rejected; nonetheless, it was not the primary cause of the ambiguous relationship between $\delta^{15}N_{POM}$ and nitrate concentration.

Second, the nitrogenous (nitrate) source of the SOJ exhibited variability. Previous studies in the northern ECS (Umezawa et al., 2014; Umezawa et al., 2021) found four nitrate sources with varying $\delta^{15}N_{NO3}$. The nitrate with high-$\delta^{15}N_{NO3}$ (8.3‰) originated from the Changjiang freshwater in July, the nitrate with low-$\delta^{15}N_{NO3}$ (2.0‰) originated from the Changjiang estuary in July, the $\delta^{15}N_{NO3}$ in the water originating from the Kuroshio is 5.5–6.0‰ in February, and that originating from atmospheric deposition is -4–0‰ (Umezawa et al., 2014; Umezawa et al., 2021). Furthermore, active nitrogen fixation occurs in the northwestern ECS (Shiozaki et al., 2010), and $\delta^{15}N_{POM}$ originating from nitrogen fixation is -2.1–0.8‰ (Minagawa and Wada, 1986). According to these results, the $\delta^{15}N_{POM}$ formed by nitrate assimilation and nitrogen fixation exhibited a wider range. In fact, in the ECS, where the TWC originates, $\delta^{15}N_{POM}$ near the surface varies widely by about -5–9‰ during summer (Gao et al., 2014) and 2–6‰ in autumn (Wu et al., 2003). Horizontal advective transport of nitrate from the ECS is the key factor for controlling primary production in the SOJ throughout the summer (Kodama et al., 2015; Kodama et al., 2017). POM originating from diverse nitrogenous sources will be mixed during the horizontal advection processes in the TWC and the ECS, and POM from the ECS is expected to flow into the SOJ with nitrate. The numerous nitrogen source contributions would obscure the $\delta^{15}N_{POM}$ and nitrate concentration in the SOJ.

Here, a simulation of the relationship between the $\delta^{15}N_{POM}$ and nitrate concentration was performed (see, the supporting material). The $\delta^{15}N_{NO3}$ was set to 0–8.3‰ (Umezawa et al., 2014; Umezawa et al., 2021), the kinetic isotope effect of nitrate assimilation to 3‰ (Sigman et al., 2009), and the initially supplied nitrate concentration to 0.05–10 µM. The $\delta^{15}N_{POM}$ was then calculated by mixing nitrate-origin POM and nitrogen-fixation-origin POM. Based on observations in the southern ECS in summer, the contribution of nitrogen fixation to nitrate assimilation in the water column was reported as 10–82% (Liu et al., 2013). $\delta^{15}N_{POM}$ produced with nitrogen fixation was set at -2.1–0.8‰ (Minagawa and Wada, 1986). The fraction of remanent nitrate was set as 0–50%. Then $\delta^{15}N_{POM}$ was calculated based on Sigman et al. (2009) with the random values in the setting ranges except for the kinetic isotope effect, the sample size was set to 500, and the relationship between $\delta^{15}N_{POM}$ and remnant nitrate concentration was evaluated. When this comparison was conducted 1000 times, the insignificant relationship between the $\delta^{15}N_{POM}$ and nitrate concentration was observed in ~70% of the simulations (Figure S2). On the other hand, when the $\delta^{15}N_{NO3}$ was adjusted to 5–6‰, the significant negative relationship between $\delta^{15}N_{POM}$ and nitrate concentration was consistently observed (Figure S2). This result supports our hypothesis that the relationship between the $\delta^{15}N_{POM}$ and nitrate concentration is disrupted by the numerous nitrogen sources.

Another distinctive feature of the SOJ was the low-$\delta^{15}N_{POM}$ designated class I, which was found only in the winter, and is characterized in the T-S diagrams. The low-$\delta^{15}N_{POM}$ was mainly observed in temperature and salinity ranges of 9.4–11.4°C and 33.877–34.038, respectively. Wagawa et al. (2020) classified this water as "upper low salinity water" (ULSW). Despite the fact that the origin of ULSW was unclear, Wagawa et al. (2020) proposed that it originated from Toyama Bay, with a less saline condition caused by the mixing with local Japanese rivers. It is reasonable to assume that the ULSW is not mixed with the saline TWC water (its salinity is ~ 34.5). Because this saline TWC water originates from the Kuroshio Current, the $\delta^{15}N_{NO3}$ of the saline TWC water is estimated to be 5.5–6.0‰ in accordance with Umezawa et al. (2014). At the same time, the $\delta^{15}N_{NO3}$ of local Japanese rivers has been estimated to be 0–2‰ (Sugimoto et al., 2019), suggesting that the POM in the ULSW may

have originated from lower $\delta^{15}N_{NO3}$-nitrate than the Kuroshio-origin nitrate, and hence the $\delta^{15}N_{POM}$ is lower than the other water masses. The horizontal distribution of ULSW is not reported, but we assumed that it was not large and confined to winter and spring based on Wagawa et al. (2020), hence the low-$\delta^{15}N_{POM}$ area would be limited to season and area. Phytoplankton bloom is another possibility for class I. $\delta^{15}N_{POM}$ rapidly declined at the start of the phytoplankton bloom phase but quickly rose with nitrate depletion (Nakatsuka et al., 1992). The class I samples were collected in February, and the phytoplankton bloom occurred at the end of March (Kodama et al., 2018; Maúre et al., 2017), but the chlorophyll-*a* concentration was not low in class I, and the C:N ratio ($6.05 \pm 0.58$ mol mol$^{-1}$) of class I (Fig. 6) did not suggest that the strong river-origin POM contribution. Therefore, the effects of phytoplankton bloom could not be rejected.

Seasonality remained significant after accounting for hydrographic conditions in the GLM. Not only nitrate concentrations but also nitrogen sources have a seasonality in the SOJ. The mixed layer deepens in the winter, and hence the nitrate originated in deep-sea water or local Japanese rivers in this season. The utilization of nitrate provided in winter occurs in spring, and the Tsushima Warm Current remains weak (Yabe et al., 2021). Kodama et al. (2015) also showed that subsurface nutrient maximum induced by the horizontal advective transport of the Tsushima Warm Current could be observed from the beginning of June. Therefore, the seasonality of $\delta^{15}N_{POM}$ may be associated with horizontal advective transport, albeit additional research is required to understand the seasonality.

## 5. Conclusions

In this study, the $\delta^{13}C_{POM}$ and $\delta^{15}N_{POM}$ values in the southern SOJ were investigated and reported for the first time. Our observations were mostly conducted in the summer, therefore our identified characteristics of $\delta^{13}C_{POM}$ and $\delta^{15}N_{POM}$ mainly reflected the characteristics of the summer of SOJ, while seasonality, except autumn, was covered by the monitoring line. There were significant seasonal variations in $\delta^{13}C_{POM}$ and $\delta^{15}N_{POM}$ in the surface mixed layer, but not below the mixed layer (at 30 m depth). Environmental variables and primary production processes adequately explained the observed $\delta^{13}C_{POM}$ value. $\delta^{13}C_{POM}$ could be estimated using our GLM and routine hydrographic observations. However, environmental variables did not adequately explain the variation in $\delta^{15}N_{POM}$. In particular, the relationship between nitrate concentration was not found. The SOJ contains different nitrogenous sources, such as atmospheric depositions and riverine inputs, and $\delta^{15}N_{POM}$ indicates that these sources are mixed and support primary production. The simulation supported that multiple nitrate sources contributed to the ambiguous relationship between $\delta^{15}N_{POM}$ and nitrate concentration. The main nitrogen source in the SOJ was not detected in our study, but the new production was dependent on the nitrate supplied from these sources. Anthropogenic nitrogen inputs were increased in the SOJ (Duan et al., 2007; Kitayama et al., 2012), and hence, the anthropogenic-nitrogen-induced production in the SOJ is expected to increase. As a result, we must evaluate the impact of "increased" production on the biogeochemical cycles and ecosystems in the SOJ.

## Code and Data availability

The stable isotope ratio of POM with environmental data used for the statistical analysis in the study is available at PANGAEA (https://doi.org/10.1594/PANGAEA.949287).

## Author contribution

TK and KN designed the experiments. All carried them out. TK and AN prepared the manuscript with contributions from all co-authors.

## Competing interests:

The authors declare that they have no conflict of interest.

## Acknowledgements

We would like to thank the captains, crews, researchers, and staff who took part in the cruises. In particular, we would like to thank Keiko Yamada for her assistance with the stable isotope analysis. We state that there is no conflict of interest, and the Japanese government granted authorization for all of observations in accordance with the Japanese legislation. This work has been funded by a research and assessment program for fisheries resources from the Fisheries Agency of Japan, grants from the Japan Fisheries Research and Education Agency, and Japan Society for the Promotion of Science grants for Taketoshi Kodama, Misato Nakae, Yosuke Igeta, and Yoichi Kogure (19K06198, 22K03729, and 23H02285).

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

**Table 1** Summary of the sampling cruises. Date, longitude, and latitude denote their ranges in the cruise, and *n* means the number of samples. SI-line indicated that the observations were conducted on the SI-line during the cruise while the collection was also conducted at the other site.

| Year | Vessel | Date | Longitude (°E) | Latitude (°N) | *n* | |
|------|--------|------|----------------|---------------|-----|---|
| 2016 | Shunyo-Maru | Jul 24–Aug 05 | 133°28'–138°20' | 35°45'–38°44' | 68 | |
| | Dai-Go Kaiyo-Maru | Sep 03–Sep 13 | 137°04'–139°04' | 37°34'–40°34' | 20 | SI-line |
| 2017 | Mizuho-Maru | Apr 21–Apr 24 | 135°20'–137°20' | 35°40'–37°40' | 7 | |
| 2018 | Shunyo-Maru | Jul 24–Aug 01 | 133°30'–138°15' | 36°00'–39°35' | 24 | |
| | Hokko-Maru | Aug 30–Sep 08 | 136°45'–139°02' | 37°35'–40°20' | 48 | SI-line |
| 2019 | Hokko-Maru | Feb 25–Feb 25 | 137°35'–138°14' | 38°08'–39°05' | 3 | SI-line |
| | Shunyo-Maru | Apr 15–Apr 17 | 136°20'–138°14' | 38°08'–41°00' | 3 | SI-line |
| | Yoko-Maru | May 23–May 25 | 135°20'–137°20' | 35°50'–37°20' | 6 | |
| | Tenyo-Maru | Jun 18–Jun 20 | 136°20'–138°13' | 38°08'–41°00' | 6 | SI-line |
| | Shunyo-Maru | Jul 13–Jul 28 | 133°30'–138°15' | 35°45'–39°35' | 42 | |
| | Hokko-Maru | Aug 30–Sep 08 | 136°49'–138°55' | 37°20'–40°22' | 46 | SI-line |
| | Tenyo-Maru | Sep 20–Sep 21 | 136°20'–138°14' | 38°08'–41°00' | 6 | SI-line |
| 2020 | Hokko-Maru | Feb 21–Feb 25 | 136°21'–138°14' | 38°08'–41°00' | 6 | SI-line |
| | Shunyo-Maru | Apr 12–Apr 14 | 136°20'–138°13' | 38°08'–41°00' | 6 | SI-line |
| | Yoko-Maru | May 24–May 27 | 135°35'–138°00' | 35°40'–37°30' | 14 | |
| | Tenyo-Maru | Jun 15–Jun 17 | 136°20'–138°13' | 38°08'–41°00' | 6 | SI-line |
| | | Jul 15–Jul 27 | 138°06'–139°50' | 37°17'–39°55' | 21 | |
| | Yoko-Maru | Aug 01–Aug 09 | 131°15'–135°45' | 35°00'–37°45' | 26 | |
| | Hokko-Maru | Aug 29–Sep 08 | 136°05'–139°25' | 37°36'–40°38' | 92 | SI-line |
| | Tenyo-Maru | Sep 22–Sep 23 | 136°20'–137°35' | 39°05'–41°00' | 4 | SI-line |
| 2021 | Hokko-Maru | Feb 22–Feb 23 | 136°20'–138°14' | 38°08'–41°00' | 6 | SI-line |
| | Shunyo-Maru | Apr 16–Apr 17 | 136°20'–138°14' | 38°08'–41°00' | 3 | SI-line |
| | Yoko-Maru | May 21–May 24 | 135°50'–138°00' | 36°00'–37°40' | 8 | |
| | Tenyo-Maru | Jun 11–Jun 13 | 136°21'–138°14' | 38°08'–41°00' | 3 | SI-line |
| | Yoko-Maru | Jul 14–Jul 19 | 133°30'–135°00' | 35°45'–37°45' | 16 | |
| | Tenyo-Maru | Jul 17–Jul 24 | 138°46'–139°50' | 38°16'–39°58' | 32 | |

# Figure

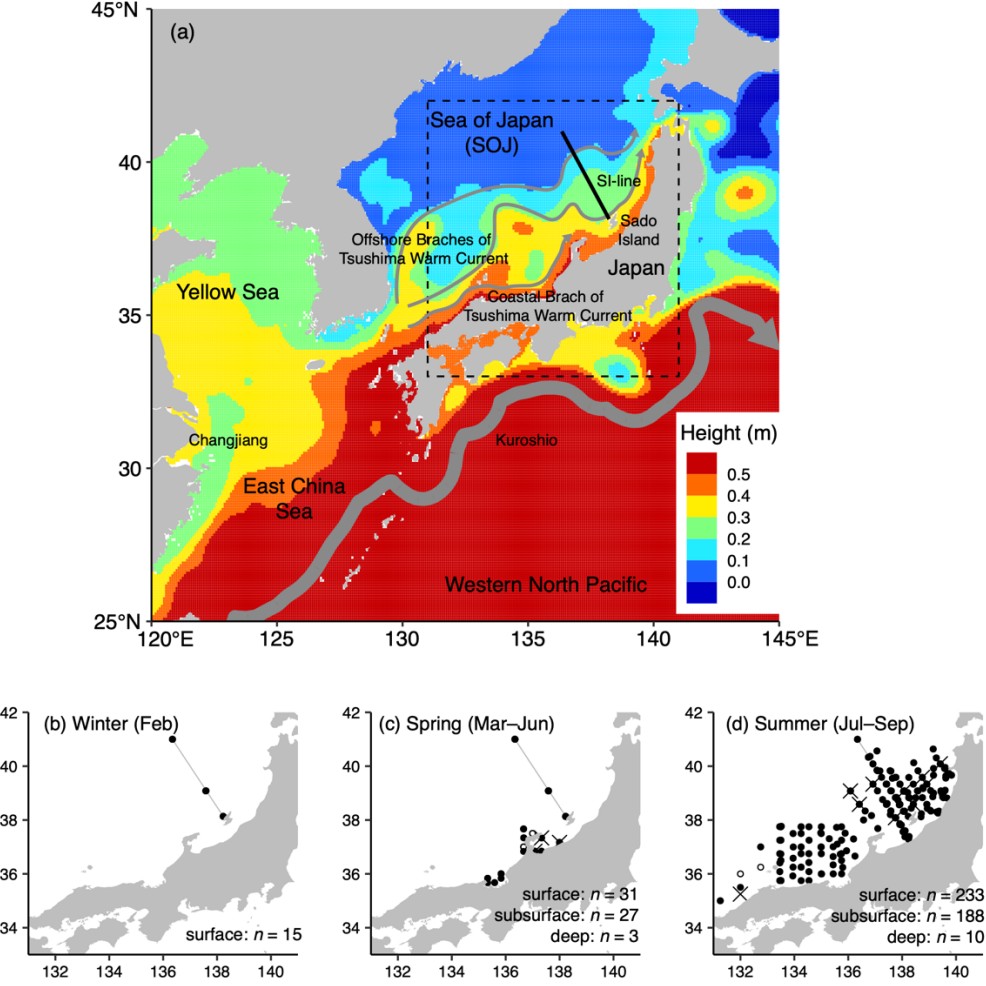

**Figure 1.** Map of the sampling stations. (a) Small-scale map of our observation area (break line square) with sea surface height and estimated ocean current positions (arrows: two offshore branches of Tsushima Warm Current, coastal branches of Tsushima Warm Current, and Kuroshio). (b) Sampling stations in winter (February), (c) in spring (March–June), and (d) in summer (July–September). The sea surface height was derived from Copernicus Marine Service Global Ocean Physics Reanalysis (GLORYS12V1, https://doi.org/10.48670/moi-00021) in August 2015. Along the SI-line [black solid line in (a) and gray line in (b) – (d)] set offshore of Sado Island, repeated observations were made. The crosses indicate the stations where samples were collected from the deep layer (75 m in spring and 100 m in summer), and open circles indicate stations where samples were collected only from the subsurface layer (20–65 m depth). In winter, all the sampling layers were assumed to be in the mixed layer, thus we did not divide them into surface and subsurface layers when collections were made at 10 m and 30 m depths. The maps were made with Natural Earth, and Geospatial Information Authority of Japan.

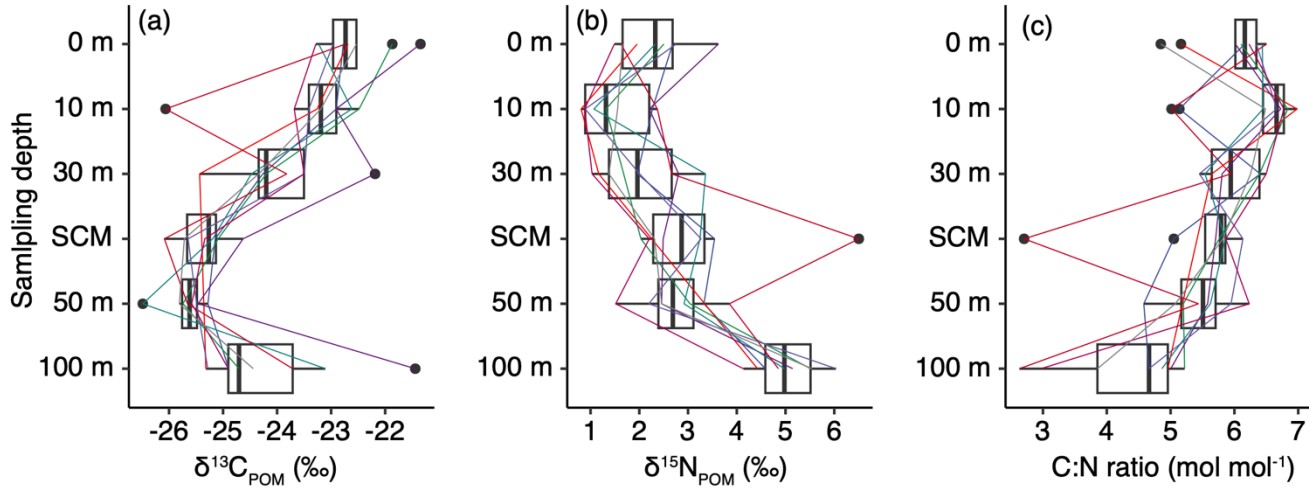

**Figure 2.** The vertical profiles of (a) $\delta^{13}C_{POM}$, (b) $\delta^{15}N_{POM}$, and (c) the C:N ratio collected at nine stations in the eastern part of the Sea of Japan during September of 2020 (cross stations in Figure 1). Lines denote the profiles of every station. Box plots show the median (vertical thick lines within boxes), upper and lower quartiles (boxes), quartile deviations (horizontal bars), and outliers (closed circles).

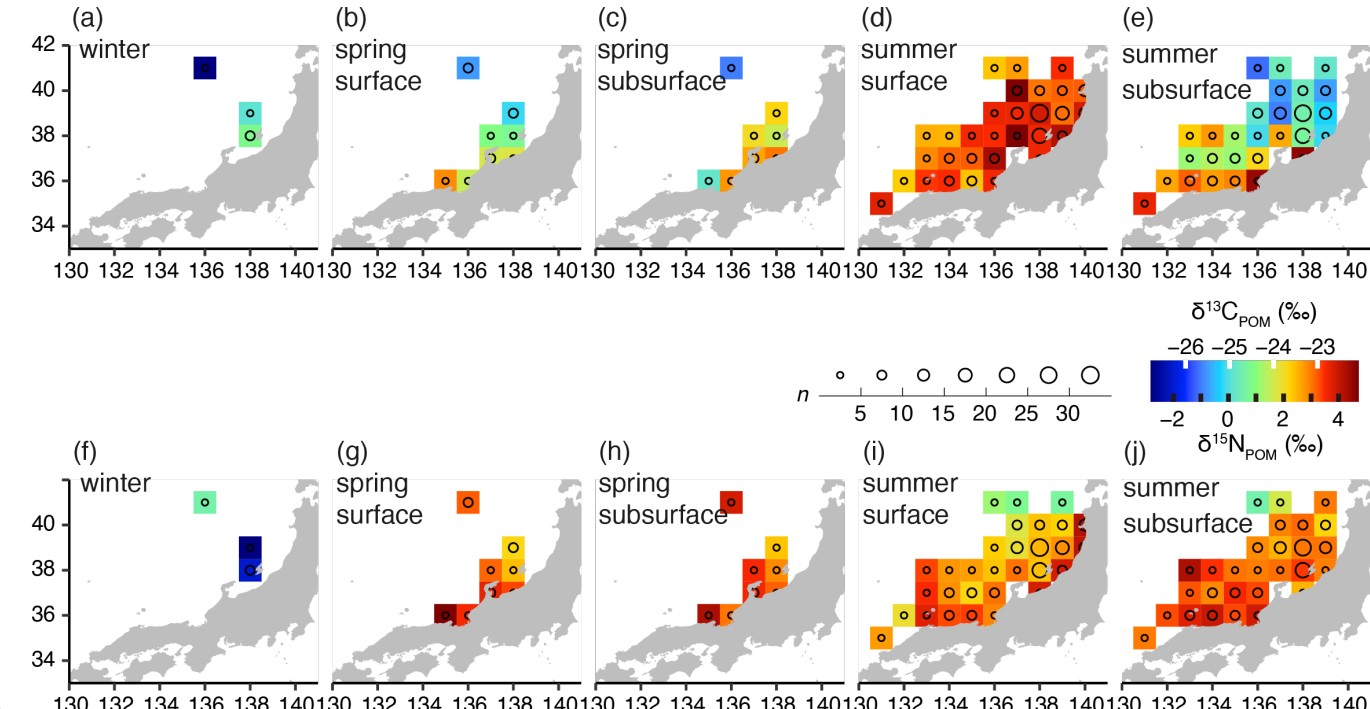

**Figure 3.** Horizontal distributions of $\delta^{13}C_{POM}$ (upper, a–e) and $\delta^{15}N_{POM}$ (lower, f–j) in the Sea of Japan (1° × 1° grid median values for five years). The (a and f) winter, (b and g) in the surface layer of spring, (c and h) in the subsurface layer of spring, (d and i) in the surface layer of summer, and (e and j) in the subsurface layer of summer. Circle sizes reflect the sample numbers used for calculating the median values. The maps were made with Natural Earth, and Geospatial Information Authority of Japan.

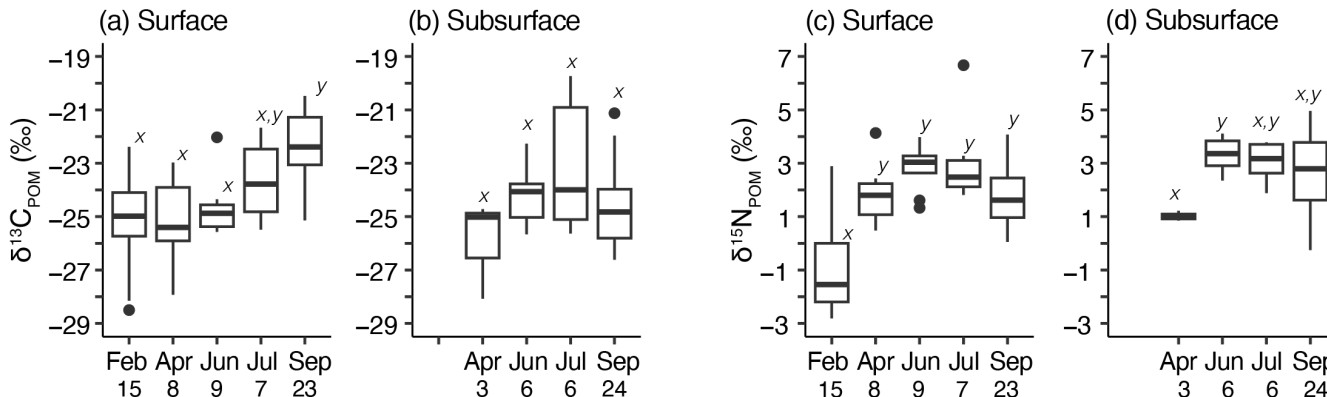

**Figure 4**. Temporal variations of $\delta^{13}C_{POM}$ (left) and $\delta^{15}N_{POM}$ (right) along the monitoring line (SI-line); (a) Monthly variation of $\delta^{13}C_{POM}$ in the surface layer, (b) monthly variation of $\delta^{13}C_{POM}$ in the subsurface layer, (c) monthly variation of $\delta^{15}N_{POM}$ in the surface layer, and (d) monthly variation of $\delta^{15}N_{POM}$ in the subsurface layer. Box plots show the mean (thick horizontal lines within boxes), standard deviation (boxes), maximum or minimum values (vertical bars), and outliers (closed circles). The lower-case italic characters near the boxes (x, and y) reflect the results of the pairwise test with Tukey–Kramer adjacent; the same character pairs showed an insignificant difference in the pair ($p > 0.05$). The pairwise test with Tukey–Kramer adjacent was not significant among any pairs of $\delta^{13}C_{POM}$ in the subsurface layer (c). The numbers just below the horizontal axis lables indicate the sample numbers.

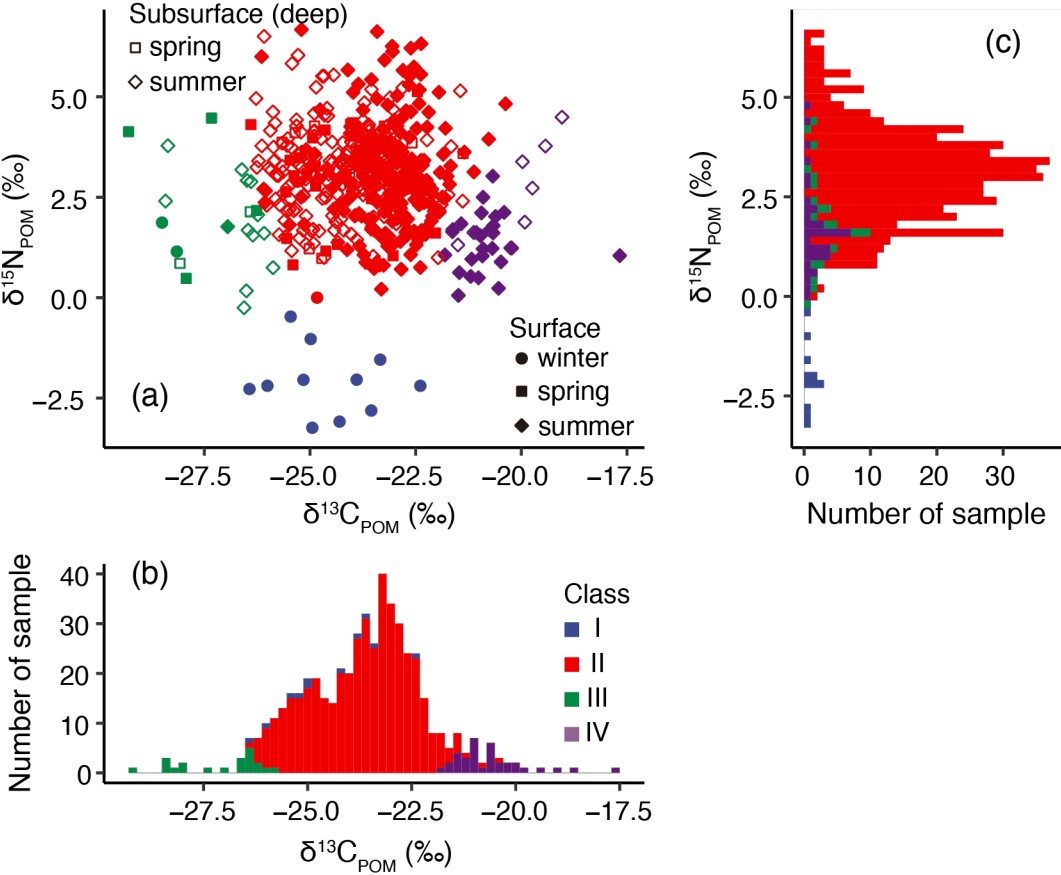

**Figure 5.** (a) Diagram between $\delta^{15}N_{POM}$ and $\delta^{13}C_{POM}$. Stacked histograms of (b) $\delta^{13}C_{POM}$ and (c) $\delta^{15}N_{POM}$ for the two-dimensional Gaussian mixed model (GMM), with colors indicating the different classes.

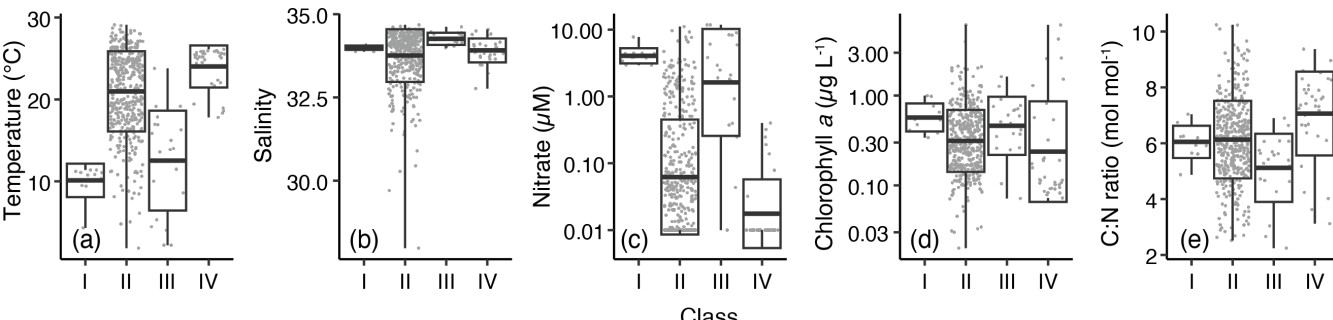

**Figure. 6**. Differences in environmental parameters among the classes divided based on $\delta^{13}C_{POM}$ and $\delta^{15}N_{POM}$. Parameters include (a) temperature, (b) salinity, (c) nitrate concentration, (d) chlorophyll-*a* concentration, and (e) C:N ratio. Box plots show the mean (thick horizontal lines within boxes), standard deviation (boxes), and maximum or minimum values (vertical bars). The small gray dots represent the raw values.

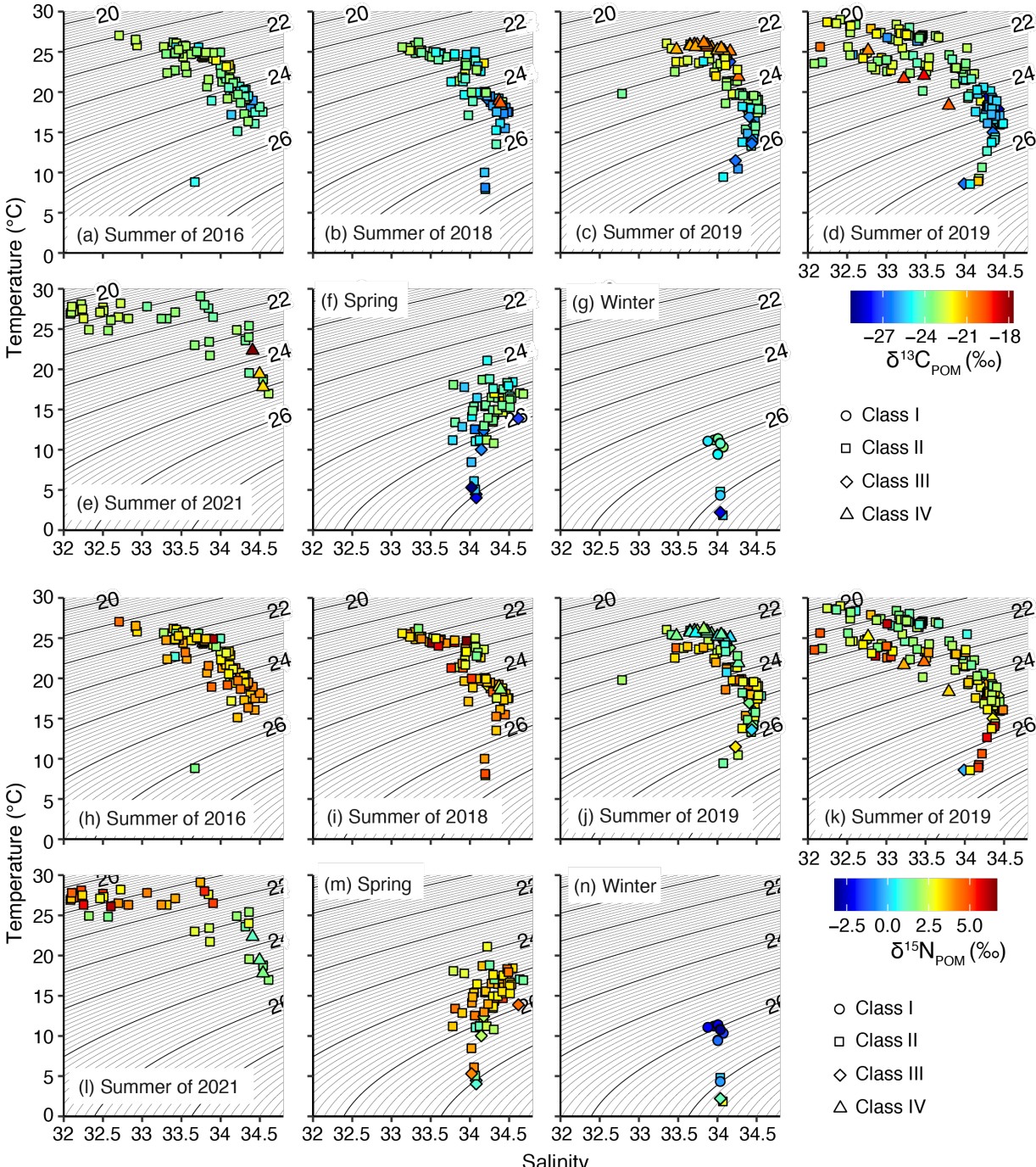

**Figure. 7.** T–S (temperature–salinity) diagram overlayed with $\delta^{13}C_{POM}$ (a–g) and $\delta^{15}N_{POM}$ (h–n). The spring (f, m) and winter (g, n) were not divided into observation years. In this diagram, salinity <32 was not plotted. The different symbols indicate different classes.

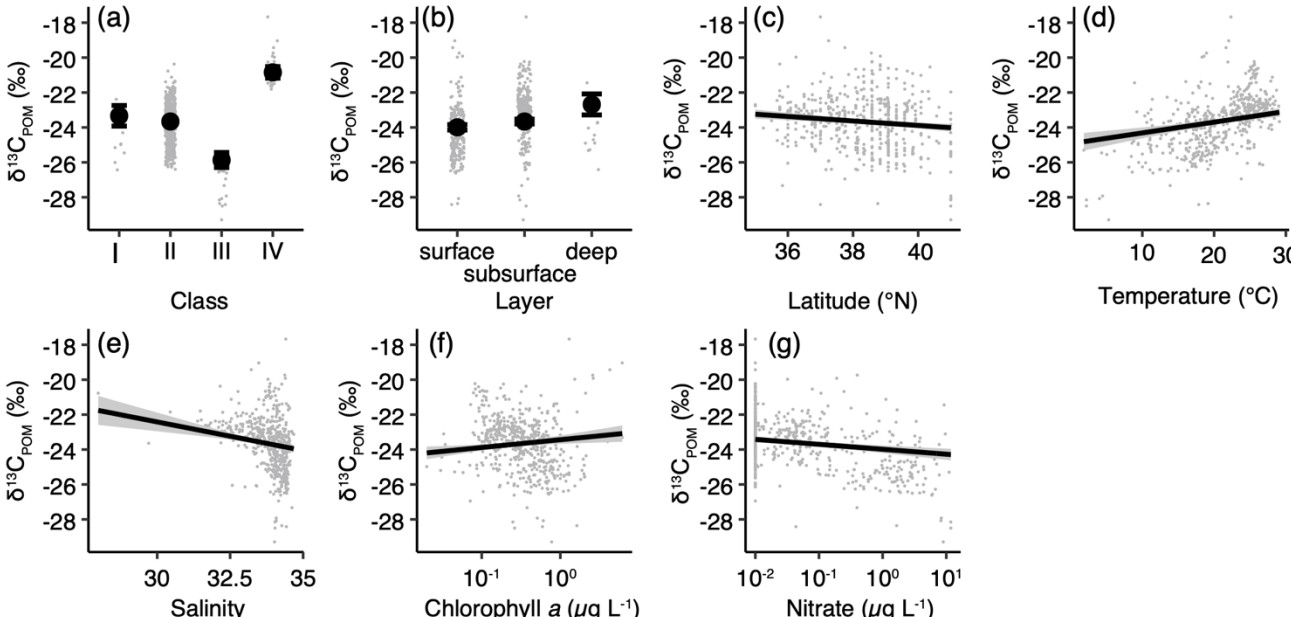

**Figure 8.** The least-square mean (lsmean) based effects of the environmental parameters in the least AIC GLM for $\delta^{13}C_{POM}$. Effect of (a) classes, (b) sampling depth, (c) latitude, (d) temperature, (e) salinity, (f) log-transformed chlorophyll-*a* concentration, and (g) log-transformed nitrate concentration. Closed circles with bars or solid lines with shadows represent the lsmeans with 95 % confidence intervals. The small gray dots represent the observation data. When necessary, the settings of the class and layer for calculation of lsmeans were class II and surface, respectively.

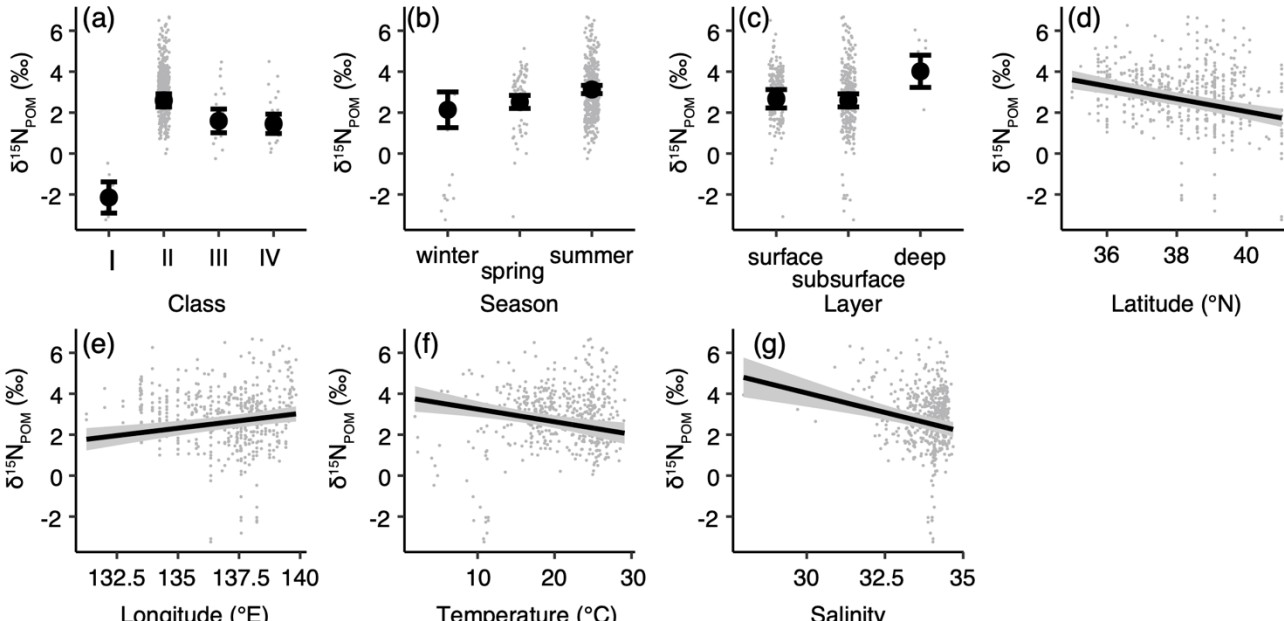

**Figure 9.** Least-square mean (lsmean) values based on the effects of the environmental parameters in the least-AIC GLM for
645 $\delta^{15}N_{POM}$. Effect of (a) classes, (b) sampling seasons, (c) sampling depth, (d) latitude, (e) longitude, (f) temperature, and (g) salinity. Closed circles with bars or solid lines with shadows represent the lsmeans with 95 % confidence intervals. The small gray dots represent the observation data. When necessary, the settings of the class, season and layer for calculation of lsmeans were class II, summer, and surface, respectively.