# Peer review of "Multiple nitrogen sources for primary production inferred from $\delta^{13}C$ and $\delta^{15}N$ in the southern Sea of Japan"

_EGUsphere, 2023_

## Referee Comment (RC1)

**Summary**

This manuscript by Kodama et al. investigated carbon and nitrogen isotope ratios of particulate organic matter (POM) in the upper water column of the southern Sea of Japan based on multiple years' observation to evaluate the effects of lateral transport of POC from the ECS and identify the nitrogen sources that support the POM production. I applaud their efforts that compiled abundant data for the first time in this region, but recommend the authors revisit these valuable data they collected and interpret them in a better way. The discussion in the present manuscript needs to be strengthened, since there are a lot of handwavy statements and speculated interpretation. I have some major concerns that should be addressed.

1. The abstract needs to be improved. One of the major goals in this work, the effects of lateral transport of POC from the ECS, was not indicated in the abstract. And why was the characteristics of class I samples emphasized here? In addition, the authors attributed variations in the $\delta^{15}N_{POM}$ to the temperature and salinity. How do these two parameters change the $\delta^{15}N_{POM}$? Do the authors mean different water masses with variable N isotope endmember that support the formation of POM? At last, their conclusion suggested multiple nitrogen sources contributing to the primary production in the SOJ, which seems to be consistent with the previous findings stated in the introduction. In this regard, the contribution of these plentiful dataset could be limited.

2. The interpretation of the $\delta^{13}C_{POM}$ variations. I suggest the authors to examine the water masses during the sampling period in the study area, for example, based on T-S diagram. This may help identify the POM sources and the sources of nitrate which is related to the $\delta^{15}N_{POM}$.

   POM with high $\delta^{13}C_{POM}$ were originated from the less saline ECS waters (Line 300-304). Since the salinity in the northeastern ECS is generally higher than 32 in summer (e.g., Kubota et al., 2015; Yang et al., 2021), it is hard to understand that the water mass with the lower end of salinity (<30) observed in this study originates from the ECS/the Changjiang diluted water. Also, when checking the spatial distribution of $\delta^{13}C_{POM}$ in Figure 3d, I do not see higher $\delta^{13}C_{POM}$ at sites close to the Tsushima Strait where the

influence of the ECS water may be higher. By contrast, higher $\delta^{13}C_{POM}$ seem to be observed in the eastern part of study area.

Line 312-314: I do not agree that the high $\delta^{13}C_{POM}$ can not be observed at sites with low-nutrient and low-Chl a. For example, in the oligotrophic oceans, e.g., the SCS basin (Liu et al., 2007), high $\delta^{13}C_{POM}$ can be detected in the surface water. In addition, considering the observed range of C/N ratio which were mainly close to the Redfield ratio, this may suggest a predominantly marine origin of POM. Liu et al. (2007) suggested that lower $\delta^{13}C_{POM}$ (theoretically down to ~ -27‰ at 100 m) in the subsurface and deep layers were due to reduced specific growth rate, which may produce larger isotope fractionation. Could this mechanism influence the spatiotemporal variations of $\delta^{13}C_{POM}$ in this study, especially for the data from the subsurface and deep waters and from the growth-limited season (i.e., winter).

3.  The interpretation of the $\delta^{15}N_{POM}$ variations. The $\delta^{15}N_{POM}$ increased with depth, which may imply that the POM degradation preferentially remove [14]N from particles, as indicated by many previous studies (e.g., Casciotti et al., 2008). In this case, the classification of $\delta^{15}N_{POM}$ which combined surface, subsurface and deep samples together would complicate the identification of nitrogen sources.

    The less saline ECS waters were suggested to be contributed to high $\delta^{15}N_{POM}$ in the SOJ (Line 334-336). Similar to the spatial pattern of $\delta^{13}C_{POM}$, in the western SOJ close to the Tsushima Strait the $\delta^{15}N_{POM}$ values were lower.

    I can not understand the simulation shown in Line 377-389, the authors need to explain in details how the various nitrogen sources change the relationship between the $\delta^{15}N_{POM}$ and nitrate concentration.

    It is interesting that such [15]N-depleted signals were observed on the POM. The authors excluded the possibility of phytoplankton growth that produces low $\delta^{15}N_{POM}$ in the nitrate-replete condition. However, the Chl a concentrations for class I were not low,

instead, seem to be highest among the four classes. Also, the authors failed to explain why the ULSW may have a very low $\delta^{15}N$ of nitrate.

At last, for the discussion on the interannual variation in $\delta^{13}C_{POM}$ and $\delta^{15}N_{POM}$, what is the message the authors intended to deliver?

Yang, L., Zhang, J. and Yang, G. P., Mixing behavior, biological and photolytic degradation of dissolved organic matter in the East China Sea and the Yellow Sea, Science of the Total Environment, 2021, 143164.

Kubota, Y., Tada, R., and Kimoto, K., Changes in East Asian summer monsoon precipitation during the Holocene deduced from a freshwater flux reconstruction of the Changjiang (Yangtze River) based on the oxygen isotope mass balance in the northern East China Sea. Climate of Past, 2015, 265-281.

Liu, K. K., S. J. Kao, H. C. Hu, W. C. Chou, G. W. Hung, and C. M. Tseng (2007), Carbon isotopic composition of suspended and sinking particulate organic matter in the northern South China Sea - From production to deposition, *Deep-Sea Research Part II*, *54*(14-15), 1504-1527.

Casciotti, K. L., T. W. Trull, D. M. Glover, and D. Davies (2008), Constraints on nitrogen cycling at the subtropical North Pacific Station ALOHA from isotopic measurements of nitrate and particulate nitrogen, *Deep Sea Research Part II: Topical Studies in Oceanography*, *55*(14-15), 1661-1672.

**Minor comments**

Line 17: original

Line 64: $\delta15N_S$?

Line 75: Sampling. I encourage the authors to add a table to summarize the sampling information.

Line 102-103: why the detection limit of nutrient was shown in a range?

Line 118: The authors did not use the international isotope standards to calibrate the data. How did they prove the precision of these data?

Line 171 and Line 185: check the significant digitals of SD. And the significant digitals of mean $\delta^{13}C$ and $\delta^{15}N$ are different.

Line 205: Explain more about the classification criterion of carbon and nitrogen isotope ratios of POM

Line 226-229: many samples had very low C/N ratio (down to 3), mainly observed at deep depth (Figure 2). Why?

Line 302: Strait

Line 303-305: I can not follow the purpose for the seasonality of estuarine $\delta^{13}C_{POM}$ mentioned here.

Line 355-356: what is the third hypothesis?

Line 375: Any evidence that supports the particles can be entrained from the ECS into the SOJ?

Line 382-383: To me, the POM pool is not fully newly produced and is not only supported by new production. So the assumed contribution of nitrogen fixation to the POM pool here, that is 10-82% (Liu et al., 2013), is too high.

Line 433-434: As mentioned, this study did not identify the main source of nitrogen in this region. The importance of anthropogenic nitrogen inputs in this region was not indicated in this study. I could agree the increasing inputs of anthropogenic nitrogen in the future, but changes in new nitrogen inputs from other sources should be considered. For instance, nitrogen fixation may be inhibited due to higher inputs of anthropogenic nitrogen. In addition, warming-induced stratification in the water column may prevent upwelled nitrate from the subsurface.

---

## Author Comment (AC1)

**Response to RC1 comments**

This manuscript by Kodama et al. investigated carbon and nitrogen isotope ratios of particulate organic matter (POM) in the upper water column of the southern Sea of Japan based on multiple years' observation to evaluate the effects of lateral transport of POC from the ECS and identify the nitrogen sources that support the POM production. I applaud their efforts that compiled abundant data for the first time in this region, but recommend the authors revisit these valuable data they collected and interpret them in a better way. The discussion in the present manuscript needs to be strengthened, since there are a lot of handwavy statements and speculated interpretation. I have some major concerns that should be addressed.

**We deeply acknowledge Reviewer #1's comments and appreciate the opportunity to respond to them. Reviewer #1's comments on our analysis helped us improve our manuscript explanation significantly.**
**During the revision process, we found that we had made a mistake in the data sets. We had accidentally put the same dates on the different cruises. We are very sorry for this careless mistake. As a result, some of the results and discussions have been revised.**
**In particular, we found that all the samples classified as class I were collected during the winter. As a result, we have deleted some of the discussion about class I.**
**On the other hand, we could have shown a more favorable conclusion if we had limited the data and used an inadequate statistical approach. However, we believe that this approach would have been dishonest. As a result, our analyses may not be sufficient for Reviewer #1. Therefore, we have revised our manuscript to address Reviewer #1's major concerns. We believe that our manuscript is now improved and better meets Reviewer #1's comments.**

1. The abstract needs to be improved.

   **We appreciate the comment.  As reviewer#1 commented, our manuscript has a gap between the aims and abstract/conclusions.  We have revised our abstract and the aims of this study as follows.**

   1) One of the major goals in this work, the effects of lateral transport of POC from the ECS, was not indicated in the abstract.

      **We appreciate this comment.  The horizontal advection of POC was removed from the aims.  We believe that the horizontal advective transport of POC from the ECS could not be evaluated in the manuscript because we did not have data in the ECS (L74).**

   2) And why was the characteristics of class I samples emphasized here?

**In Class I, this very low δ¹⁵N value was one of our new findings in this study. To emphasize our consideration, we added the short explanation "whose δ¹⁵N_POM showed an outlier of total data sets" (L16).**

3) In addition, the authors attributed variations in the δ¹⁵NPOM to the temperature and salinity. How do these two parameters change the δ¹⁵NPOM? Do the authors mean different water masses with variable N isotope endmember that support the formation of POM?

**We appreciate this comment. Yes, we considered, and so added the short description as "which suggested that the δ¹⁵N of source nitrogen is different among the water masses" (L20–21).**

4) At last, their conclusion suggested multiple nitrogen sources contributing to the primary production in the SOJ, which seems to be consistent with the previous findings stated in the introduction. In this regard, the contribution of these plentiful dataset could be limited.

**We are sorry but could not exactly understand the meaning. Previous studies have not reported on the contribution of nitrogen sources to production. We believe that this was the progress of our study, so we keep it.**

2. The interpretation of the δ¹³CPOM variations.

   **We appreciate the comment very much. We revised our manuscript followed by the comments. However, we believe that reviewer#1 might have misunderstood our results and discussions in few cases.**

1) I suggest the authors to examine the water masses during the sampling period in the study area, for example, based on T-S diagram. This may help identify the POM sources and the sources of nitrate which is related to the δ¹⁵NPOM.

   **We deeply appreciate this comment. We agree that the T-S diagram is a useful way to identify water masses, as already written in the manuscript**

[Figure]

**(L156), and the results are shown in Figure 7a.  However, we re-created the T-S diagram as shown in the upper: we put δ13C and δ15N values on the T-S diagram based on reviewer#1 comment.  The T-S diagrams did not clearly show the characteristics of δ13C and δ15N.  These figures must be more informative than Figure 7, so we revised Figure 7.**

2) POM with high $\delta^{13}C_{POM}$ were originated from the less saline ECS waters (Line 300-304). Since the salinity in the northeastern ECS is generally higher than 32 in summer (e.g., Kubota et al., 2015; Yang et al., 2021), it is hard to understand that the water mass with the lower end of salinity (<30) observed in this study originates from the ECS/the Changjiang diluted water.

**We are sorry but we cannot agree with this comment because many studies reported less-saline (<32) water originating in Changjiang in the northeastern part of the ECS and the Tsushima Strait (Morimoto et al., 2009; Umezawa et al., 2014; Kodama et al., 2017).  The introduced studies are a snapshot observation (Yang et al., 2021) or a sediment observation (Kubota et al., 2015).  Kubota et al. (2015) showed the horizontal distribution of salinity, but references describing the salinity data were not included in the reference lists.  So, we did not revise here.**

3) Also, when checking the spatial distribution of $\delta^{13}C_{POM}$ in Figure 3d, I do not see higher $\delta^{13}C_{POM}$ at sites close to the Tsushima Strait where the influence of the ECS water may be higher. By contrast, higher $\delta^{13}C_{POM}$ seem to be observed in the eastern part of study area.

**We are afraid that review #1 may have misunderstood our results and discussion.  As we had described, δ13C is not only controlled by salinity, but also by temperature and chlorophyll *a* concentration.  This means that δ13C was not always associated with the water mass and depended on the *in-situ* conditions as well.**

4) Line 312-314: I do not agree that the high $\delta^{13}C_{POM}$ can not be observed at sites with low- nutrient and low-Chl a. For example, in the oligotrophic oceans, e.g., the SCS basin (Liu et al., 2007), high $\delta^{13}C_{POM}$ can be detected in the surface water. In addition, considering the observed range of C/N ratio which were mainly close to the Redfield ratio, this may suggest a predominantly marine origin of POM. Liu et al. (2007) suggested that lower $\delta^{13}C_{POM}$ (theoretically down to ~ -27‰ at 100 m) in the subsurface and deep layers were due to reduced specific growth rate, which may produce larger isotope fractionation. Could this mechanism influence the spatiotemporal variations of $\delta^{13}C_{POM}$ in this study, especially for the data from the subsurface and deep waters and from the growth-limited season (i.e., winter).

**We are sorry but we cannot understand this comment.  We believe the reviewer**

misunderstand here.  We did not describe as "high $\delta^{13}C_{POM}$ can not be observed at sites with low-nutrient and low-Chl a," we described as "the reason for high $\delta^{13}C_{POM}$ values was unidentified."  In addition, we discussed the high $\delta^{13}C_{POM}$ values here and not the low $\delta^{13}C_{POM}$.  We have already discussed that low $\delta^{13}C_{POM}$ may depend on limited growth at L303-309.  However, we have not referred to Liu et al. (2007), and we have referred to this in the revised manuscript.

3.  The interpretation of the $\delta^{15}NPOM$ variations.

1) The $\delta^{15}NPOM$ increased with depth, which may imply that the POM degradation preferentially remove $^{14}N$ from particles, as indicated by many previous studies (e.g., Casciotti et al., 2008). In this case, the classification of $\delta^{15}NPOM$ which combined surface, subsurface and deep samples together would complicate the identification of nitrogen sources.

We appreciate reviewer#1 comment.  The subsurface layer in our study was in the euphotic layer, and Casciotti et al. (2008) showed that degradation occurred in a deeper layer (several hundred meters in depth).  Thus, in our case, the degradation processes are hardly considered, except in the deep layer, as shown in the GLM approach.  In addition, the classification is not only for the identification of the $\delta^{15}N_{POM}$ origin, but also for statistical analyses.  In our data sets, we could not set the statistical distribution of $\delta^{15}N_{POM}$ as a normal distribution when we did not conduct this classification.  Even a simple linear model was inappropriate when the data distribution was normal.  Therefore, this approach is considered the most appropriate method for statistical analysis.

2) The less saline ECS waters were suggested to be contributed to high $\delta^{15}NPOM$ in the SOJ (Line 334-336). Similar to the spatial pattern of $\delta^{13}CPOM$, in the western SOJ close to the Tsushima Strait the $\delta^{15}NPOM$ values were lower.

Again, the spatial distribution of $\delta^{15}N_{POM}$ was not associated with distance from the source, and other factors also contributed.

3) I can not understand the simulation shown in Line 377-389, the authors need to explain in details how the various nitrogen sources change the relationship between the $\delta^{15}NPOM$ and nitrate concentration.

We appreciate this comment.  We considered that our descriptions were not enough at all.  Thus, we prepared supporting material for this paragraph.  In the supporting material, we added the figure shown below.  To simplify, when $\delta^{15}N_{NO3}$ varies widely, the significant negative relationship between $\delta^{15}N_{POM}$ and nitrate concentration could not be observed in some cases. When the $\delta^{15}N_{NO3}$ had a wider

range, the $\delta^{15}N_{POM}$ also showed a wider range as well. Since there are multiple nitrogen sources in the Sea of Japan, the $\delta^{15}N_{POM}$ showed a wide range.

[Figure]

**Figure S2. The results of the simulations when the $\delta^{15}N_{NO3}$ varies $0 - 8.3‰$ and the contribution of nitrogen fixation ($f_{N2}$) was 10–82% (a and b), those $\delta^{15}N_{NO3}$ varies 5 – 6‰ and $f_{N2}$ was 10–82% (c and d), and those $\delta^{15}N_{NO3}$ varies $0 - 8.3‰$ and $f_{N2}$ was 1.9–5.8% (e and f). (a), (c) and (e) denote the representative result of their relationship of each simulation, and (b), (d) and (f) were the histograms of the $p$-values of simulations repeated 1000 times. The open bar in (b), (d) and (f) denote the $p$-values were ≥0.05 (significant), and the closed bars denote $p$-values were >0.05.**

4) It is interesting that such $^{15}N$-depleted signals were observed on the POM. The authors excluded the possibility of phytoplankton growth that produces low $\delta^{15}NPOM$ in the nitrate-replete condition. However, the Chl a concentrations for class I were not low, instead, seem to be highest among the four classes. Also, the authors failed to explain why the ULSW may have a very low δ15N of nitrate.

**We appreciate this comment. We are sorry, but we cannot understand why the reviewer commented on the chlorophyll *a* concentration. In our study, we could not find a significant relationship between δ15N$_{POM}$ and chlorophyll *a* concentration; thus, whether chlorophyll *a* concentration is high or low is not important. In addition, we already explained why the ULSW might have a very low δ15N of nitrate in the manuscript. It was because the nitrate originating from the Japanese local river is low (L386-390).**

4.  At last, for the discussion on the interannual variation in $\delta^{13}C_{POM}$ and $\delta^{15}N_{POM}$, what is the message the authors intended to deliver?

    **We agree that interannual discussion is not necessary. Since our manuscript is long, we cut interannual variations in the revised manuscript.**

**Minor comments**

Line 17: original

**We revised as "temperature and salinity originating in Japanese local rivers" (L17).**

Line 64: $\delta^{15N}$s?

**We thanked the comment. "$\delta^{15}N$ values" (L65).**

Line 75: Sampling. I encourage the authors to add a table to summarize the sampling information.

**We thanked this comment. We show it as Table 1.**

Line 102-103: why the detection limit of nutrient was shown in a range?

**It was because the detection limit varied among the measurements. We revised (L100).**

Line 118: The authors did not use the international isotope standards to calibrate the data. How did they prove the precision of these data?

**L-alanine (Shoko Science) was possible to treat as the international reference material. We added L116–L117.**

Line 171 and Line 185: check the significant digitals of SD. And the significant digitals of mean $\delta13C$ and $\delta15N$ are different.

**The significant digits of standard deviation are very complex. We did not uniform the digits.**

We uniform the one place of decimal.  The difference between δ13C and δ15N was correct. The accuracy of them was one place of decimal.

Line 205: Explain more about the classification criterion of carbon and nitrogen isotope ratios of POM

**The Gaussian mixed model was determined the number of clusters only based on the Bayesian information criterion (BIC) (L137).  So, there is no classification criterion as such a hierarchic clustering analysis.**

Line 226-229: many samples had very low C/N ratio (down to 3), mainly observed at deep depth (Figure 2). Why?

**We are sorry but we have no idea about this.  Martiny et al. 2013 reported that a low C:N ratio of POM is observed in the ~100 m depth in the global data sets while the mean value is elevated.  Xu et al 2021 reported that regeneration of C:N ratio is different among the depth. Since our profile data was very limited, we could not why it occurred.  In addition, only two of nine samples recorded a low C/N ratio.**

Line 302: Strait

**Sorry for the careless mistake (L297).**

Line 303-305: I can not follow the purpose for the seasonality of estuarine δ¹³CPOM mentioned here.

**We considered this description would be an over-discussion.  We removed (L300).**

Line 355-356: what is the third hypothesis?

**Sorry, we forgot to revise as two hypotheses (L345).**

Line 375: Any evidence that supports the particles can be entrained from the ECS into the SOJ?

**We cannot clearly understand this comment.  Particles must be transported with water.**

Line 382-383: To me, the POM pool is not fully newly produced and is not only supported by new production. So the assumed contribution of nitrogen fixation to the POM pool here, that is 10-82% (Liu et al., 2013), is too high.

**This is an important point.  Our description is not enough.  The details were added in the supporting material, and we also added the analysis, which contribution was set as low.  At first, we know all the primary production is not new production.  The regenerated nutrient is contributed to primary production as well.  However, the regenerated nutrient was supplied as the new nutrients or via nitrogen fixation at first.  Thus, the contribution of $N_2$ fixation to new production not to primary production is important.  However, nitrogen fixation contribution must be lower in the spring, as reviewer #1 suggested.  So we added the analysis.   We revised the main text (L367–379) as well as Supporting Materials.**

Line 433-434: As mentioned, this study did not identify the main source of nitrogen in this region. The importance of anthropogenic nitrogen inputs in this region was not indicated in this study. I could agree the increasing inputs of anthropogenic nitrogen in the future, but changes in new nitrogen inputs from other sources should be considered. For instance, nitrogen fixation may be inhibited due to higher inputs of anthropogenic nitrogen. In addition, warming-induced stratification in the water column may prevent upwelled nitrate from the subsurface.

**We appreciate this comment.  The review #1 comment is correct and we easily described this sentence.  We revised the sentence as "the anthropogenic-nitrogen-induced production in the SOJ is expected to increase. As a result, we must evaluate the impact of "increased" production on the biogeochemical cycles and ecosystems in the SOJ" (L414–417).**

---

## Author Comment (AC2)

**Response to Reviewer#2**

This study presents the spatial and temporal distribution of carbon and nitrogen stable isotope signals in euphotic particulate organic matter (POM), which is primarily made up of autotrophic phytoplankton, in the southern Sea of Japan. Elucidating the sources of POM and the potential biogeochemical processes that modify POM isotopic characteristics in the marginal sea, which is impacted by multiple currents and complex biogeochemistry, is interesting and important. In my opinion, the manuscript's overall structure and analyses are satisfactory. However, I concur with Referee 1's remarks and would like to see more in-depth discussions of the data. Here are a few additional comments regarding the analyses.

**We appreciate Reviewer#2 comments very much and also apricate to revise our manuscript. We revised manuscript based on Reviewe#2 comments as well as Reviewer#1 comment. We found some careless mistakes in my data sets. We revised the sentence for the class I samples due to our careless mistake.**

The data points in class II are significantly more abundant than the other classes and exhibit high variations in both stable isotope signals and environmental conditions. However, the majority of data points in this class were collected during the summer, which may render seasonality insignificant. If only data from this class are used, may it be possible to establish correlations between $\delta 13C/\delta 15N$ and environmental factors, particularly Chl-a and nitrate, that could explain the variation in $\delta 13C/\delta 15N$?

**We appreciate reviewer #2's comments. We agree with their opinion and have already done what they suggested. We noted in our manuscript that we also analyzed the data for only the summer. When we limited the data to class II, we observed some differences. The $\delta^{13}C$ was explained by season, temperature, salinity, and nitrate concentration. The trend was the same as with the full data set. The sampling layer, latitude, and chlorophyll *a* concentration were not selected. In the case of $\delta^{15}N$, only sampling depth and salinity were selected as the explanatory variables. The model of $\delta^{15}N$ was poorly explained by these two parameters (the deviance explained was less than 10%). We believe these results are more difficult to explain, and considering the deviance explained, these models are inadequate compared to the model we described in the manuscript. We carefully considered the data and**

**used appropriate statistical methods. We believe that our results are valid and that our conclusions are justified.**

In reference to the first comment, classes III and IV mainly consist of summer and a few spring samplings, but the δ13C values in these groups differ significantly from class II. The proposed explanation that high and low δ13C is related to the strength of photosynthesis is accepted, albeit inferred indirectly from nitrate depletion and high C:N ratios. Although there is no data available to confirm it, the stable isotope difference between nitrate and POM may reflect the depletion of nitrate caused by phytoplankton activity. Also, please provide relevant literature that links high phytoplankton C:N ratios with high photosynthetic C assimilation.

**We appreciate the comment. We added the description about the C/N ratio as follows: "In addition, the C/N ratio was not selected in the GLM but had a positive relationship with $\delta^{13}C_{POM}$ (Figure S1). Tanioka and Matsumoto (2020) reported that C/N ratio elevates with the increase of light based on the meta-analysis, supporting the active photosynthesis elevates both C/N ratio and $\delta^{13}C_{POM}$" (L278–281).**

I am unclear about why it was mentioned that "class IV samples were mostly from the surface layer during the summer, but the surface layer samples during summer were not classified as the class III (Lines 310-311)." Does this suggest that biological processes differ between the surface and deep waters? For instance, in deep water where nitrate levels are high but light intensity is low, and thus phytoplankton photosynthesis is limited, resulting in less significant δ13C fractionation.

**We are sorry for our confusing descriptions. We want to say that class III and class IV water was present in different season and/or layer. Since reviewer #2 could not understand, we revised the order of the sentences and removed the unclear sentences, and we emphasize the photosynthesis (L303–309).**

I am curious as to why the nitrate-σt biplot (Figure 7b) was investigated. Given the high frequency of below-detection limit NO3-, this biplot may not be particularly useful for

identifying water masses and grouping POM stable isotope signals.

**We thank this comment.   Now we agree with this comment, and thus we only show the T-S diagram.   The T-S diagram was revised based on reviewer#1 comment (Figure 7).**

[Figure]

Minor comments:

Lines 393-394: USLW should be ULSW in the two sentences.

**We appreciate this comment.   We revised (L383 and 385)**

Figure 9: According to Fig. 5a, Class 1 includes almost exclusively winter data. The δ15N distribution should be similar in 9a and 9b. However, the average δ15N shown in in 9b is much higher, which obviously deviated from the data distribution. Please check the codes creating this subplot and fix it.

**We appreciate this comment.   To calculate the lsmean values, the categorical values were set as the majority ones.   And thus, the $\delta^{15}N$ value in winter was calculated as class II and surface.   This is why the winter lsmean value looks high in Fig 9b.   In addition, there are some plots behind the circles and bars.   We added the description in the caption of Fig 8 and 9.**

---

## Referee Report (RR1)

I am happy to see the revised version of the manuscript. It has become clearer and more succinct. The responses to my comments are satisfying and I can understand what the authors aimed to present. I have minor comments on the text.

Line 180: would it be Fig. 3f-j?

Line 278: delete "utilization".

Line 283: The negative relationship between "$\delta^{13}C_{POM}$ and" salinity…

Lines 301-305: Let me clarify the logic. Does this mean that "in class III, there is deep mixing of water column and thus high nitrate; however, light intensity was low in winter and spring (it seemed that most points in class III were spring and winter samplings) and deep mixing brought phytoplankton to deeper water, which lowered the phytoplankton activity and $\delta^{13}C_{POM}$"?

To the comment 1-(3) from Reviewer #1, I guess what the reviewer meant is that you mentioned that previous studies (Umezawa et al. 2021; Umezawa et al. 2014) already found that the primary production is supported by multiple identified N sources in the northeastern ECS and what this research found does not contribute much to new findings. However, your research area is not exactly the same as previous studies (though SOJ receives influences from the ECS). I think this is still a good dataset to explore the relationships between N sources and its effect on ocean production.

And the comment 2-(3) from Reviewer #1 may infer to this statement "Additionally, the elevated $\delta^{13}C_{POM}$ may be due to sediment resuspension in the Changjiang estuary…" (Lines 285-290), and "These results suggest that POM with high $\delta^{13}C_{POM}$ is transported into the SOJ and influences the spatiotemporal variation of $\delta^{13}C_{POM}$ in SOJ,…" (Lines 295-299). Indeed, this is confusing. Maybe you can mention that the salinity is not the main factor affecting $\delta^{13}C_{POM}$ in the end of this paragraph to emphasize that chl-a and phytoplankton photosynthesis matter.

---

## Editor Decision (ED1)

Reviewer #1

I have read the revised manuscript by Kodama et al. and am pleased to see that the authors have taken time and effort to improve it. I have some minor concerns as follows.

1) As mentioned in the Introduction (Lines 65-68), previous studies have found that the nitrate supplied to the SOJ surface water originates from Kuroshio, ECS, Changjiang diluted waters, atmosphere, and local deep waters. In my opinion, these findings suggest multiple nitrogen sources that can support phytoplankton growth. Can authors clarify whether the present data support the previous results or provide any new understanding?

2) I still have a question about the explanation of the 15N-depleted signals of POM for class I. The authors excluded the possibility of phytoplankton growth that produces low $\delta^{15}N_{POM}$ in the nitrate-replete condition. I do understand that the authors claimed no relationship between $\delta^{15}N_{POM}$ and Chl a, which implied that $\delta^{15}N_{POM}$ may not be determined by Chl a. As I stated before, the Chl a concentrations for class I were not low; instead, they seemed to be the highest among the four classes, while the authors argue that the bloom generally occurs in spring. Higher Chl a concentrations indicated phytoplankton growth. Moreover, the $\delta^{13}C_{POM}$ values for class I were moderate and C/N ratios for class I were close to 6, suggesting that the marine-origin POM significantly contributed to the bulk POM. These results together demonstrate that the in situ and/or remote produced POM by phytoplankton may not be fully ruled out.
* * *
Reviewer #2

I am happy to see the revised version of the manuscript. It has become clearer and more succinct. The responses to my comments are satisfying and I can understand what the authors aimed to present. I have minor comments on the text.
Line 180: would it be Fig. 3f-j?
Line 278: delete "utilization".
Line 283: The negative relationship between "d13CPOM and" salinity...
Lines 301-305: Let me clarify the logic. Does this mean that "in class III, there is deep mixing of water column and thus high nitrate; however, light intensity was low in winter and spring (it seemed that most points in class III were spring and winter samplings) and deep mixing brought phytoplankton to deeper water, which lowered the phytoplankton activity and d13CPOM"?

To the comment 1-(3) from Reviewer #1, I guess what the reviewer meant is that you mentioned that previous studies (Umezawa et al. 2021; Umezawa et al. 2014) already found that the primary production is supported by multiple identified N sources in the northeastern ECS and what this research found does not contribute much to new findings. However, your research area is not exactly the same as previous studies (though SOJ receives influences from the ECS). I think this is still a good dataset to explore the relationships between N sources and its effect on ocean production.

And the comment 2-(3) from Reviewer #1 may infer to this statement "Additionally, the elevated d13CPOM may be due to sediment resuspension in the Changjiang estuary..." (Lines 285-290), and "These results suggest that POM with high d13CPOM is transported into the SOJ and influences the spatiotemporal variation of d13CPOM in SOJ,..." (Lines 295-299). Indeed, this is confusing. Maybe you can mention that the salinity is not the main factor affecting d13CPOM in the end of this paragraph to emphasize that chl-a and phytoplankton photosynthesis matter.

---

## Author Response (AR2)

**Dear Prof. Koji Suzuki**

The two referees and I were pleased to find your significant improvements in the revised manuscript. However, the two referees still have some concerns about your discussion (see the attachment). So please make a further revised manuscript with point-by-point responses to the referees' comments.

**We appreciate the opportunity to revise our manuscript again. We revised our manuscript based on all the comments from two reviewers and checked the manuscript again by ourselves. We believe our manuscript is now prepared for publishing from Biogeosciences.**

**Response to Reviewer #1**

I have read the revised manuscript by Kodama et al. and am pleased to see that the authors have taken time and effort to improve it. I have some minor concerns as follows.

**We appreciate Reviewer#1 again for his/her careful reading and valuable comments. We also thank the opportunity to revise our manuscript. We revised our manuscript based on the comments.**

1) As mentioned in the Introduction (Lines 65-68), previous studies have found that the nitrate supplied to the SOJ surface water originates from Kuroshio, ECS, Changjiang diluted waters, atmosphere, and local deep waters. In my opinion, these findings suggest multiple nitrogen sources that can support phytoplankton growth. Can authors clarify whether the present data support the previous results or provide any new understanding?

**We appreciate the comments. We are sorry that our previous revision is insufficient. Our descriptions which were pointed were confusing. To clarify our description, we arranged the sentences (L60–71).**

2) I still have a question about the explanation of the 15N-depleted signals of POM for class I. The authors excluded the possibility of phytoplankton growth that produces low $\delta15N_{POM}$ in the nitrate-replete condition. I do understand that the authors claimed no relationship between $\delta15N_{POM}$ and Chl a, which implied that $\delta15N_{POM}$ may not be determined by Chl a. As I stated before, the Chl a concentrations for class I were not low; instead, they seemed to be the highest among the four classes, while the authors argue that the bloom generally occurs in spring. Higher Chl a concentrations indicated phytoplankton growth. Moreover, the $\delta13C_{POM}$ values for class I were moderate and C/ N ratios for class I were close to 6, suggesting that the marine-origin POM significantly contributed to the bulk POM. These results together demonstrate that the in situ and/or remote produced POM by phytoplankton may not be fully ruled out.

**We appreciate the comments again. We reconsidered Reviewer#1's comment on this phenomenon, and we recognized this comment is more reasonable than our discussion. The phytoplankton carbon and nitrogen assimilation in winter is not clearly reported in the Sea of Japan, and we could not reject the Reviewer#1's comments. We revised the discussion and added the descriptions in the discussion part (L394–398).**

**Response to Reviewer #2**

I am happy to see the revised version of the manuscript. It has become clearer and more succinct. The responses to my comments are satisfying and I can understand what the authors aimed to present. I have minor comments on the text.

**We appreciate the careful and kind comments from Reviewer#2. We revised the commented parts based on the comments.**

Line 180: would it be Fig. 3f-j?
**We appreciate the comment. We revised (L181)**

Line 278: delete "utilization".
**We revised as suggested (L279).**

Line 283: The negative relationship between "d13CPOM and" salinity...
**We added (L284).**

Lines 301-305: Let me clarify the logic. Does this mean that "in class III, there is deep mixing of water column and thus high nitrate; however, light intensity was low in winter and spring (it seemed that most points in class III were spring and winter samplings) and deep mixing brought phytoplankton to deeper water, which lowered the phytoplankton activity and d13CPOM"?

**We thank the suggestion. Most of Reviewer#2 comprehension is correct, but we note not only deep water mixing, but also the sampling depth lowered d13CPOM due to the low light intensity and low phytoplankton activity. We clearly state this (L306–308).**

To the comment 1-(3) from Reviewer #1, I guess what the reviewer meant is that you mentioned that previous studies (Umezawa et al. 2021; Umezawa et al. 2014) already found that the primary production is supported by multiple identified N sources in the northeastern ECS and what this research found does not contribute much to new findings. However, your research area is not exactly the same as previous studies (though SOJ receives influences from the ECS). I think this is still a good dataset to explore the relationships between N sources and its effect on ocean production.

**We appreciate this comment. Thankfully, Reviewer#1 commented it us again. We revised the sentences following your comments (L60–71).**

And the comment 2-(3) from Reviewer #1 may infer to this statement "Additionally, the elevated d13CPOM may be due to sediment resuspension in the Changjiang estuary..." (Lines 285-290), and "These results suggest that POM with high d13CPOM is transported into the SOJ and influences the

spatiotemporal variation of d13CPOM in SOJ,..." (Lines 295-299). Indeed, this is confusing. Maybe you can mention that the salinity is not the main factor affecting d13CPOM in the end of this paragraph to emphasize that chl-a and phytoplankton photosynthesis matter.

**We also appreciate this comment. As suggested by Reviewer#2, we revised the paragraph and reordered the sentences. We cannot reject the effect of salinity, so we did not mention "salinity is not the main factor affecting d13CPOM" (L393–396).**

---

## Author Response (AR3)

**Dear Prof. Koji Suzuki**

The last referee was delighted to find the significant improvements in your manuscript but still provided a few minor comments below:

There is a small mistake in line 194. I think the subplot cited in this sentence should be 4c. And there is a repetitive description of subsurface DeltaC and Fig. 4b in line 194 (it has been described in lines 191-192).

**We greatly thank your handling and effort in our manuscript.  We believe our manuscript is ready for acceptance.**
**We revised the pointed sentences (L194).  We thank the comments very much.**

**Taketoshi Kodama**

---

## Author Response (AR4)

Our colour schemes used in the maps and charts were revised when they were inappropriate for readers with colour vision deficiencies based on the remarks from the preceding review file validation.